# Mapping potential risks for the transmission of spotted fever rickettsiosis: The case study from the Rio de Janeiro state, Brazil

Cláudio Manuel Rodrigues[1]*, Francisco Dourado[2], Daniel Savignon Marinho[3], Gilberto Salles Gazêta[4], Lena Geise[5]

1 Oswaldo Cruz Foundation, Center for Technological Development in Health (Fiocruz / CDTS), Rio de Janeiro, Rio de Janeiro, Brazil, 2 Department of Applied Geology, Center for Disaster Study and Research (UERJ/CEPEDES), University of Rio de Janeiro State, Rio de Janeiro, Rio de Janeiro, Brazil, 3 Department of Epidemiology and Quantitative Methods in Health (Fiocruz/ENSP/DEMQS), Oswaldo Cruz Foundation, Sergio Arouca National School of Public Health, Rio de Janeiro, Rio de Janeiro, Brazil, 4 Oswaldo Cruz Foundation, Oswaldo Cruz Institute, National Reference Laboratory for Rickettsial Vectors (Fiocruz / IOC / LIRN), Rio de Janeiro, Rio de Janeiro, Brazil, 5 Department of Zoology, Institute of Biology, University of Rio de Janeiro State, (UERJ / IBRAG/ DZ), Rio De Janeiro, Rio de Janeiro, Brazil

* 1tenclaudio@gmail.com, claudio.rodrigues@cdts.fiocruz.br

**Data Availability Statement:** All relevant data are within the paper and its Supporting information files.

## Abstract

Spotted fever rickettsiosis is a zoonosis transmitted by ticks, having a varied clinical course that can lead to death if not managed properly. In Brazil it is more commonly observed in the Southeast, being an emerging public health problem. Hazard mapping models are common in different areas of knowledge, including public health, as a way of inferring reality and seeking to reduce or prevent damage. The aim of this study is to offer a spatial heuristic methodology for assessing the potential risk of transmission of spotted fever in the Rio de Janeiro state, located in the southeastern region of Brazil. For this, we used geospatial tools associated with eco-epidemiological data related to the clinical profile of the disease. The results achieved were substantially encouraging, considering that there are territories with greater or lesser expectation of risk for spotted fever in the study area. We observed that there are important distinctions between the two rickettsiosis scenarios in the same geographic space and that the areas where there is a greater potential risk of contracting rickettsiosis coincide with the administrative regions that concentrated the cases of hospitalization and deaths from the disease, concluding that the scenery found are relevant to the case series for the disease and that the planning of surveillance actions can gain in quality if the use of this spatial analysis tool is incorporated into the routine of local health management.

## Introduction

Spotted fever rickettsiosis is an infectious disease characterized by nonspecific fever with a varied clinical course that can quickly lead the patient to death if not properly managed [1, 2]. In Brazil it is transmitted by some species of ticks, with emphasis on the genus *Amblyomma*, which contributes to the spread of bacteria of the genus *Rickettsia* among animal hosts and,

**Funding:** The author(s) received no specific funding for this work.

**Competing interests:** The authors have declared that no competing interests exist.

also, human. Among the possible etiological agents of spotted fever, *Rickettsia rickettsii* is related to the most severe cases, commonly called Brazilian spotted fever (BSF), while the other specie—*Rickettsia parkeri* strain Atlantic Rainforest—is related to mild cases, recognized in the specialized literature as spotted fever (SF) [2, 3].

As stated by literature, it is possible to relate some species of ticks to certain animal hosts, according to opportunity and preference, in addition to the peculiarities of the ecological cycles related to them [1, 4]. Three tick species of the *Amblyomma* genus (*A. sculptum*, *A. aureolatum* and *A. ovale*), as well as the species *Rhipicephalus sanguineus*, notably related to the parasitism of domestic dogs [5] and the recent urbanization process of the disease in Brazil [6–8] for this study because they have greater importance, both in ecological and epidemiological aspects [9], for the construction of a potential risk map model for the transmission of spotted fever rickettsiosis in the territory of Rio de Janeiro state, located in the Southeast region from Brazil, which concentrates the occurrence of cases, mainly during spring and summer [10]. Among the main animal hosts related to the transmission of rickettsiosis in the Brazilian territory, the capybaras (*Hydrochoerus hydrochaeris*), equines (*Equus* spp.) and, more recently, domestic dogs (*Canis lupus familiaris*) are listed [7, 11].

The hazard mapping is commonly used in various areas of knowledge, such as industry [12] or environmental management [13]. For health surveillance activities, some experiments can be checked [14, 15], though with very few applications on the concept of potential risk related directly to issues referred to tick-borne diseases. Rare exception, an article mentions the participatory mapping risk for spread of Lyme disease by analyzing account self-filled questionnaires for people who have had some contact with ticks associated with the use of GIS tools [16], and, more specifically, in Brazil, we observed the use of the concept of risk to analyze the situation of rickettsiosis in the state of Paraná [17] and, more recently, the mapping of brazilian spotted fever rickettsiosis through potential distribution techniques of species involved in enzootic and epizootic cycles of disease [18].

The first article [16] brings a concern for the health of the collective in view of the increase in cases of Lyme borreliosis in the Netherlands. The option of developing a methodology linked to citizen science for monitoring tick bites for a decade led to the formulation of a potential risk model for human exposure to ticks, seeking to associate the biological activity of the vector (hazard) with the activities of human beings' outdoor leisure activities and, in this way, quantify the degree of exposure. The second article [17] deals with a methodology for determining a risk area due to blood sampling from horses and canines positive for *Rickettsia rickettsii* and *Rickettsia parkeri* through the indirect immunofluorescence reaction associated with the preparation of risk probability maps by the indicatrix kriging technique. The results, despite being below the parameters of endemicity of rickettsiosis in Brazil, indicate the presence of biotic factors compatible with the maintenance of the enzootic cycle of the disease in the studied territory. The third article [18] is a study of Ecological Niche Modeling (ENM) which seeks to predict the potential distribution of the etiological agent (*Rickettsia rickettsii*), its main vectors (*Amblyomma sculptum* and *Amblyomma dubitatum*) and of certain hosts (*Hydrochoerus hydrochaeris*, *Didelphis aurita* and *Didelphis marsupialis*) of BSF rickettsiosis in the national territory. For this purpose, records of the occurrence of the vectors and hosts involved were used, as well as data from cases confirmed by the brazilian epidemiological surveillance service. As a result, a great coincidence was verified between cases of the human disease with areas suitable for a better ecological relationship between the respective vectors and hosts, reflecting the capacity of the models to anticipate the distribution of BSF rickettsiosis cases in the national territory.

We used geospatial data in vector data format related to physical, environmental, ecological and epidemiological aspects of spotted fever rickettsiosis, which combined contributed to the

classification of the potential risk for disease transmission. We evaluated the possibilities of vector interaction, more specifically of ticks of the genus *Amblyomma* (*A. sculptum*, *A. aureolatum* and *A. ovale*) and of *Rhipicephalus sanguineus* with three animal hosts (capybaras, horses and domestic dogs) related to cycles of ecological aspects of the disease in Brazil. In addition, data related to the occurrence cases, as well as hospitalizations and deaths confirmed for spotted fever rickettsiosis by the official flow of the Brazilian Health Surveillance Service were used to validate the results.

Dantas *et al.* (2001) emphasize the need to jointly analyze the various variables that define a geobiophysical system, outlining a true mosaic of natural landscapes. Therefore, an accurate observation of human action on these same natural landscapes is inevitable to forge an analysis of the set of geographic landscapes that will delimit the fundamental units of analysis for territorial planning. According to the authors, in a geoecological approach it is possible to observe the magnitude of environmental impacts that can vary depending on the nature, intensity and extent of human interventions and the amplitude of alteration previously imposed on the landscape, which consequently leads to degradation of the environment physical, loss of biodiversity and reduction of the quality of life of the human population [19].

Understanding the dynamics of land has always been important of human beings, and it is not uncommon to have, since antiquity, references to the relationship between nature and human activities. Acknowledging the practices related to land use is of great relevance to guarantee a character of sustainability in the face of environmental, social and economic issues that are customarily raised in debates on how to produce food and wealth for the population while focusing on sustainable development. The verification of economic losses resulting from the development model is an important practice in the study of land use. In this way, it is not just about identifying and recognizing possible changes resulting from the appropriation of the territory, but also accounting for the remaining natural heritage and closely following these changes so that it is possible to assess the transformations, whether positive or negative, in the near future [20].

The objective of this study is to offer a spatial heuristic methodology to assess the potential risk of spotted fever transmission in the Rio de Janeiro state with the development of a potential risk map aimed at supporting the decision of local Health managers to promote surveillance actions of tick-borne diseases.

## Materials and methods

### Study area and geospatial data

Rio de Janeiro state is located in the Southeast of Brazil (Fig 1). With an area of approximately 43,750 km$^2$ and an estimated population of 18 million people, is the third smallest unit of federation. In contrast, has the highest population density in Brazil with a population density in the range of 365 inhabitants/km$^2$. It has 92 municipalities and borders Minas Gerais, Espírito Santo, São Paulo, in addition to the Atlantic Ocean. In 2010, the Human Development Index (HDI) was calculated at 0.761 and the Infant Mortality Rate is estimated at 13.16 deaths per 1000 live births [21].

By definition, geoenvironmental domains represent morphostructures that relate to remarkable events, which are responsible for the current arrangement of the relief and for the less mutable characteristics of the landscape. From the perspective of our study, the geoenvironmental domains refer to a larger taxon, compatible with regions with the same geological and geomorphological characteristics that group the same habitats and their respective faunal and floristic communities [19].

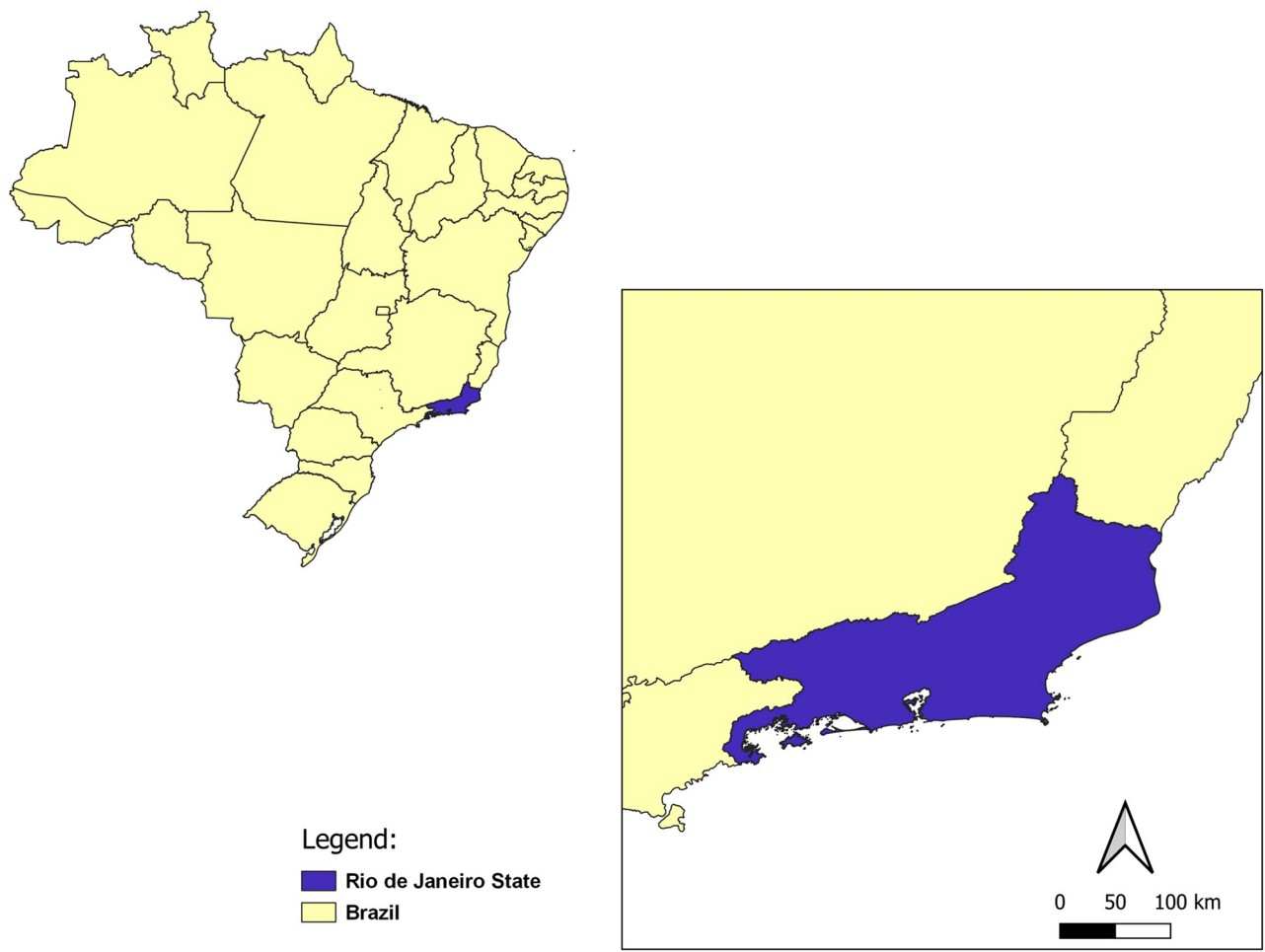

**Fig 1. Geographical representation of the study area—Rio de Janeiro state, Brazil.** Political Division. Territorial division by Federation Units. Rio de Janeiro state, Brazil. Adapted from an open access shapefile available at <https://inde.gov.br/AreaDownload#>, access Jan 28 2022 and Metadata ID: 3bba887e-6cb1-4a7d-83bd-9906b832d81a.

The geoenvironmental data used in this work are the result of a detailed study that describes the various domains and geoenvironmental units, encouraging the organization of a geoenvironmental map of the study area by the Serviço Geológico do Brasil (CPRM—Geological Survey of Brazil) [19]. The classification referring to Coverage and Land Use was compiled from the Technical Manual for Land Use of the Instituto Brasileiro de Geografia e Estatística (IBGE—Brazilian Institute of Geography and Statistics) [20], adapted for the classes of greatest interest for the study. The spatial representation of both categories is shown in Fig 2. The geographical coordinates of spatial data were converted to the SIRGAS 2000 datum before the production process of maps and illustrations. As for the right to use the vector files (shapefiles) in the production of the analyzes of this study, we obtained permission for unrestricted use, including adaptations or editions that were necessary, and publication under the Common Attribution License (CCAL) CC BY 4.0 by the respective governmental organizations that make them available on public platforms (https://www.inde.gov.br/).

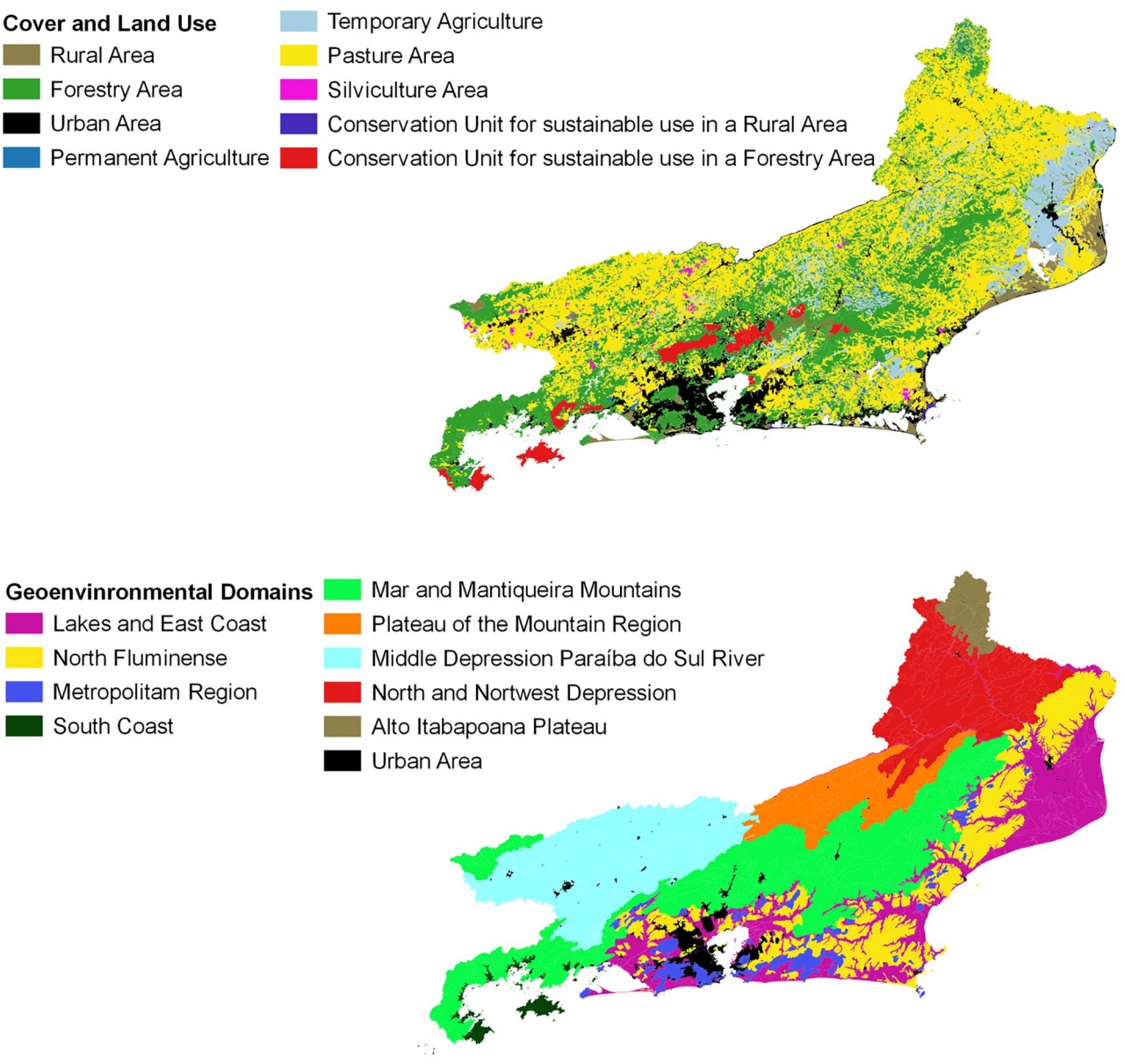

**Fig 2. Classes of cover and land use (IBGE) and geoenvironmental domains (CPRM) to Rio de Janeiro state, Brazil.** Classes of Cover and Land Use, adapted from an open access shapefile available at <https://inde.gov.br/AreaDownload#>, access Jan 28 2022, Metadata ID: 2c2589d2-e0c4-4900-9a62-870f2994b1fd; Geoenvironmental Domains, adapted from an open access shapefile available at <http://www.cprm.gov.br/publique/Gestao-Territorial/Geologia%2C-Meio-Ambiente-e-Saude/Projeto-Rio-de-Janeiro-3498.html>, access Jan 28 2022; and respective colours scales.

## Methodological proposal

Understanding that the potential risk for the transmission of spotted fever rickettsiosis (PRSF) can be demonstrated by the relationship between vectors (v) and hosts (h) projected in the geographical space, we formulated a methodological proposal. Aiming to ensure diversity of opinions in the study of the multifaceted eco-epidemiological process that maintains rickettsial etiological agents in enzootic or epizootic activity in the Rio de Janeiro state, we chose to organize a quantitative decision model with researchers and professionals from the Environmental Health Surveillance, in which we seek to achieve the best consensus.

Therefore, we would need to establish the two terms of the equation: the presence of the vector (v) would be calculated by the weights that would represent mathematically the possibility of having each species of tick according to the description of the geoenvironmental classes (φgeo) and the presence of the host (h) would be calculated by the weights that would represent mathematically the possibility of having each species of vertebrate animal in the selected cover classes and land use (φland).

Thus, to materialize the relationship between vectors (v) and hosts (h), we would develop a formula that applies to the ecological relationships between ticks (vg) and vertebrate animals (hl) in terms of geoenvironmental domains (φgeo) and the classes of cover and land use (φland) designed in cartographic space, measuring, at the end, the gradation for these relationships to occur in geographic space:

$$PR_{SF}[v, h] = \varphi_{geo}\left(v_g\right) X \varphi_{land}(h_l)$$

## Expert panels

A quantitative decision model, which was as intuitive as possible to identify the potential risks of transmission of spotted fever rickettsiosis, would be necessary to offer results that would enable decision-making by local Health Surveillance managers. Our option was to organize an expert panel [22, 23] to synthesize opinions from different points of view and point out possible uncertainties in their analysis, in order to find a balance or consensus between possibly contradictory arguments between scientific disciplines. Therefore, it was necessary to structure the problem in question, systematize antecedents so that experts could judge based on the uncertainties and argumentative strength of each discipline, carry out the panel sessions and, in the report, present the consolidated results to be plotted on maps of potential risk. Recognizing that there would be a chance that the discussion between experts would reveal contradictory ideas and results, it was understood that to ensure consensus, the figure of the referee would be necessary to organize the analysis processes, communicate clearly and concisely the decisions by consensus to the experts and, finally, write the final report.

The invited experts assigned relative weights using an intuitive numerical scale, with values ranging from 1 to 10, according to the potential relation of each of the vectors of importance in the transmission of spotted fever rickettsiosis in Rio de Janeiro state with different animal hosts (Table 1). This primary weighting process was obtained from the empirical knowledge

**Table 1. Expert panel weighted results for the ecological relationship between spotted fever rickettsiosis vectors and hosts in Rio de Janeiro state, Brazil.**

| Scenery | 1 | 2 | 3 | 4 | 5 |
|---|---|---|---|---|---|
| **Rickettsiosis diseases** | **BSF** | **BSF** | **BSF** | **SF** | **SF** |
| Dog | 1 | 1 | 2 | 7 | 1 |
| Capybara | 3 | 0 | 0 | 1 | 0 |
| Horse | 4 | 0 | 0 | 1 | 0 |
| *Amblyomma sculptum* | 7 | 0 | 0 | 0 | 0 |
| *Amblyomma aureolatum* | 0 | 1 | 0 | 0 | 0 |
| *Amblyomma ovale* | 0 | 0 | 0 | 9 | 0 |
| *Rhipicephalus sanguineus* | 0 | 0 | 2 | 0 | 1 |

BSF—Brazilian Spotted Fever and SF—Spotted Fever, Dog—*Canis familiaris*, Capybara—*Hydrochoerus hydrochaeris* and Horse—*Equus spp*.

common to 16 experts invited from universities and research centers, in addition to Public Health management in Brazil (S1 Fig and S1 Table).

Initially, a distinction was made between diseases transmitted by rickettsial agents according to the clinical status of the patients. It is recognized that the bacterium *Rickettsia rickettsii* has a very high lethality profile and, in Rio de Janeiro state, it is responsible for cases with hospitalization and death in patients affected by the disease called Brazilian Spotted Fever [24, 25]. The other rickettsiosis, with a milder clinical course, sometimes oligosymptomatic, are called Spotted Fever [26, 27].

Some scenarios were stipulated based on the relationship between vectors and hosts. Those scenarios related to Brazilian Spotted Fever (BSF) and Spotted Fever (SF) rickettsiosis received weights distributed by the consulted experts. A review of indexed scientific literature indicates that tick species of the genus *Amblyomma* (*Amblyomma sculptum* and *Amblyomma aureolatum*) have a closer relationship with BSF, while *Amblyomma ovale* maintains a strong relationship with SF [10, 28]. The tick commonly found parasitizing domestic dogs (*Rhipicephalus sanguineus*) has the potential to transmit bacteria of both symptom severity profiles [11].

A proxy was stipulated to represent possible animal hosts of ticks that are related to urban, rural and wild areas, in addition to their ecotones. In this way, it was understood that domestic dogs, horses and capybaras would be used in our study because they are easily sighted in Rio de Janeiro state and have fundamental importance in the enzootic cycle of both diseases.

The individual ecological relationship for each vector and host was measured using numerical indicators called "score v" and "score h", respectively. The higher the score, the greater the relationship of these animals with the environment of the territory occupied by Rio de Janeiro state. From these indicators it is possible to produce an indicator of ecological interaction between vectors and hosts, called "score vh", which is nothing more than the multiplication between the indicators "score v" and "score h". Finally, it is possible to produce a "final score", which translates to the ecological relationship between vectors and hosts or "score vh" and the respective disease is strong or weak when applied to the study area. The "final score" is the mathematical representation of the opportunity for each "vector-host" ecological interaction to produce one of the possible rickettsiosis diseases in Rio de Janeiro state. The sum of the "final score" results of each studied vector must reach 100 (or 1) for each rickettsiosis disease previously classified as BSF or SF (S2 Table).

In a second moment, four specialists from different areas of scientific knowledge (ecology, epidemiology, geography and biology) were recruited to develop a panel on the possible implications that anthropic action, explained by coverage and land use, has on the maintenance of possible vertebrate hosts and the relationship of geoenvironmental domains with the presence of important tick species for the maintenance of enzootic cycles of rickettsiosis and possible zoonotic spillovers that may occur in Rio de Janeiro state. Weights were assigned to each class of geographic information layers according to the potential of each vector or host to be seen in the respective locations in the study area (S3 and S4 Tables). Ticks, depending on the species, have peculiar characteristics that differentiate them from each other (*e.g.*, eating habits; degree of insolation, humidity and temperature; vegetation type; altitudinal range and anthropophilia), which reinforces their better or worse adaptation to the different geoenvironmental domains of the study area [1, 10, 11]. Hosts, on the other hand, are directly impacted by human development, explained by land use actions, with the rampant urbanization that has been taking place in Rio de Janeiro state in recent years [25, 29]. We assume that we use an intuitive simple linear scale with a constant gradation of five cardinal elements, known as a five-point Likert-Type Scale, to give scope to the experts' answers regarding the relationships between each animal species, vector or host, with the abiotic variables represented by the coverage / land use classes and geoenvironmental domains, respectively. Thus, the weights could

**Table 2. Weights relating coverage classes of coverage and land use of Rio de Janeiro state with potential hosts.**

| Code | Class of coverage and land use | Dog | Capybara | Equine |
|------|-------------------------------|-----|----------|--------|
| 11 | Urban area | 12 | 4 | 7 |
| 23 | Pasture area | 10 | 5 | 7 |
| 22 | Permanent agriculture | 9 | 4 | 6 |
| 21 | Temporary agriculture | 9 | 4 | 6 |
| 24 | Silviculture area | 9 | 4 | 6 |
| 322 | CU for sustainable use in Rural area | 14 | 5 | 7 |
| 312 | CU for sustainable use in Forest area | 15 | 4 | 6 |
| 32 | Rural area | 12 | 3 | 6 |
| 31 | Forest area | 12 | 3 | 3 |

CU = Conservation Units, Dog—*Canis familiaris*, Capybara—*Hydrochoerus hydrochaeris* and Horse—*Equus spp.*

range from 1 to 25, depending on whether the possibility of this animals occurring in a given environment was very low or almost unlikely (1x1 = 1) or very high (5x5 = 25). The final score was obtained by the simple average of the values offered by the experts, with the necessary approximation so that there were no decimal places in the results provided by the Panel (Tables 2 and 3).

## Geographic information systems operations

The expert's scores for the four vectors and the three hosts listed were linearly allocated in a table, comprising the five rickettsial diseases with ecological probability of occurring in the territory of Rio de Janeiro state (Table 1), as well as the relationship of vertebrate hosts with coverage and land use classes (Table 2) and ticks with the respective geoenvironmental domains and subdomains (Table 3) previously selected and that compose the study area. Subsequently, using ArcGIS software, we developed algebraic geoprocessing using one of the possible cartographic modeling techniques, in which georeferenced planes are combined in the same cartographic system. We chose to model vector data due to the ease of acquiring shapefiles in institutional open access repositories of the Brazilian government. We perform "overlay" operations using an intersection tool between data layers. Thus, a new data layer was created, corresponding, in the database, to columns with the respective weights assigned to each

**Table 3. Weights relating geoenvironmental domains in Rio de Janeiro state with potential vectors.**

| Code | Geoenvironmental Domain | Asculp | Aaureo | Aovale | Rsang |
|------|------------------------|--------|--------|--------|-------|
| 1 | Metropolitan Region | 11 | 3 | 4 | 8 |
| 2 | Lakes and East Coast | 15 | 3 | 4 | 8 |
| 3 | North Fluminense | 13 | 3 | 4 | 8 |
| 4 | South Coast | 14 | 3 | 7 | 8 |
| 5 | Mar and Mantiqueira Mountains | 13 | 6 | 3 | 8 |
| 6 | Plateau of the Mountain Region | 11 | 5 | 3 | 8 |
| 7 | Middle Depression Paraíba do Sul | 13 | 3 | 3 | 8 |
| 8 | North and Northwest Depression | 13 | 3 | 3 | 8 |
| 9 | Alto Itabapoana Plateau | 13 | 3 | 3 | 8 |
| 10 | Urban Area | 14 | 3 | 3 | 9 |

Asculp—*Amblyomma sculptum*; Aaureo—*Amblyomma aureolatum*; Aovale—*Amblyomma ovale*; and Rsang—*Rhipicephalus sanguineus*

**Table 4. Weights according to the specificity of sceneries studied for the transmission of spotted fever rickettsiosis.**

| Scenery | Disease | Dog | Cap | Hor | Asculp | Aaureo | Aovale | Rsang | SUM |
|---|---|---|---|---|---|---|---|---|---|
| AT | - | 1 | 1 | 1 | 1 | 1 | 1 | 1 | 7 |
| 1 | BSF | 1 | 3 | 4 | 7 | 0 | 0 | 0 | 15 |
| 2 | BSF | 1 | 0 | 0 | 0 | 1 | 0 | 0 | 2 |
| 3 | BSF | 2 | 0 | 0 | 0 | 0 | 0 | 2 | 4 |
| A | BSF | 4 | 3 | 4 | 7 | 1 | 0 | 2 | 21 |
| 4 | SF | 7 | 1 | 1 | 0 | 0 | 9 | 0 | 18 |
| 5 | SF | 1 | 0 | 0 | 0 | 0 | 0 | 1 | 2 |
| B | SF | 8 | 1 | 1 | 0 | 0 | 9 | 1 | 20 |

Dog—Canis familiaris, Cap—Hydrochoerus hydrochaeris, Hor—Equus spp., Asculp—Amblyomma sculptum, Aaureo—Amblyomma aureolatum, Aovale—Amblyomma ovale and Rsang—Rhipicephalus sanguineus. SUM—Total weights, Scenery AT (Aggregate Total)—Symptom severity scenery of entity compatible with spotted fever rickettsiosis, regardless of the etiological agent involved, considering all possibilities of host / vector interaction, Scenery 1—BSF Scenery considering the ecological relationship between the vector Amblyomma sculptum and its possible hosts (domestic dog, capybara and horse), Scenery 2—BSF Scenery considering the ecological relationship between the vector Amblyomma aureolatum and its main host, the domestic dog, Scenery 3—BSF Scenery considering the ecological relationship between the vector Rhipicephalus sanguineus and its main host, the domestic dog, Scenery A—BSF Scenery considering all possibilities of host / vector interaction, Scenery 4—Scenery of SF considering the ecological relationship between the vector Amblyomma ovale and its possible hosts (domestic dog, capybara and horse), Scenery 5—SF Scenery considering the ecological relationship between the vector Rhipicephalus sanguineus and its main host, the domestic dog, and Scenery B—SF Scenery considering all possibilities of host / vector interaction.

intersection polygon between vectors (ticks) and geoenvironmental domains or subdomains and between animal hosts and classes of coverage and use of the land. The sums of the information plans that resulted in the final value of each operation were carried out. Then, different sceneries were evaluated regardless of the classification adopted for rickettsiosis disease (BSF or SF). A Total Aggregate Scenery (TA), covering all vectors and hosts with the same weight (1) and independent of the symptom severity classification, was followed by Sceneries 1 to 5, which identify possible ecological relationships between invertebrate vectors and vertebrate hosts related to brazilian spotted fever (BSF) and spotted fever (SF). Finally, a subsidiary scenery for each symptom severity presentation of the disease was stipulated based on the sum of the weights for Sceneries 1 to 3 (Scenery A or BSF) and 4 and 5 (Scenery B or SF). Originally, the sum of the weights for each variable in each of the sceneries ranged between 9 and 200 points, with each scenery presenting different ranges of variation, which would make it impossible to directly compare them. The solution adopted to establish a comparison between the sceneries was to normalize each scenery according to the weighted average of the sum of the weights, which led to normalized results between 2 and 21 points, as detailed in Table 4 and Figs 3 and 4.

## Epidemiological validation

The rickettsiosis can be difficult to diagnose clinically, especially if the patient does not report previous contact with ticks or animals, with an initial phase that is confused with several pathologies of abrupt onset and that cause nonspecific fever, myalgia, headache, nausea and emesis. It has multisystem characteristics, and may present a variable clinical course, from classic cases of maculopapular exanthema, with centripetal evolution and predominance in the lower limbs, to atypical forms without the presence of exanthema [2, 3, 10, 25].

Laboratory diagnosis can be performed by collecting paired clinical samples of blood, serum or plasma from suspected patients, and the indirect immunofluorescence reaction is considered the gold standard for rickettsiosis in Brazil. This method is established by the

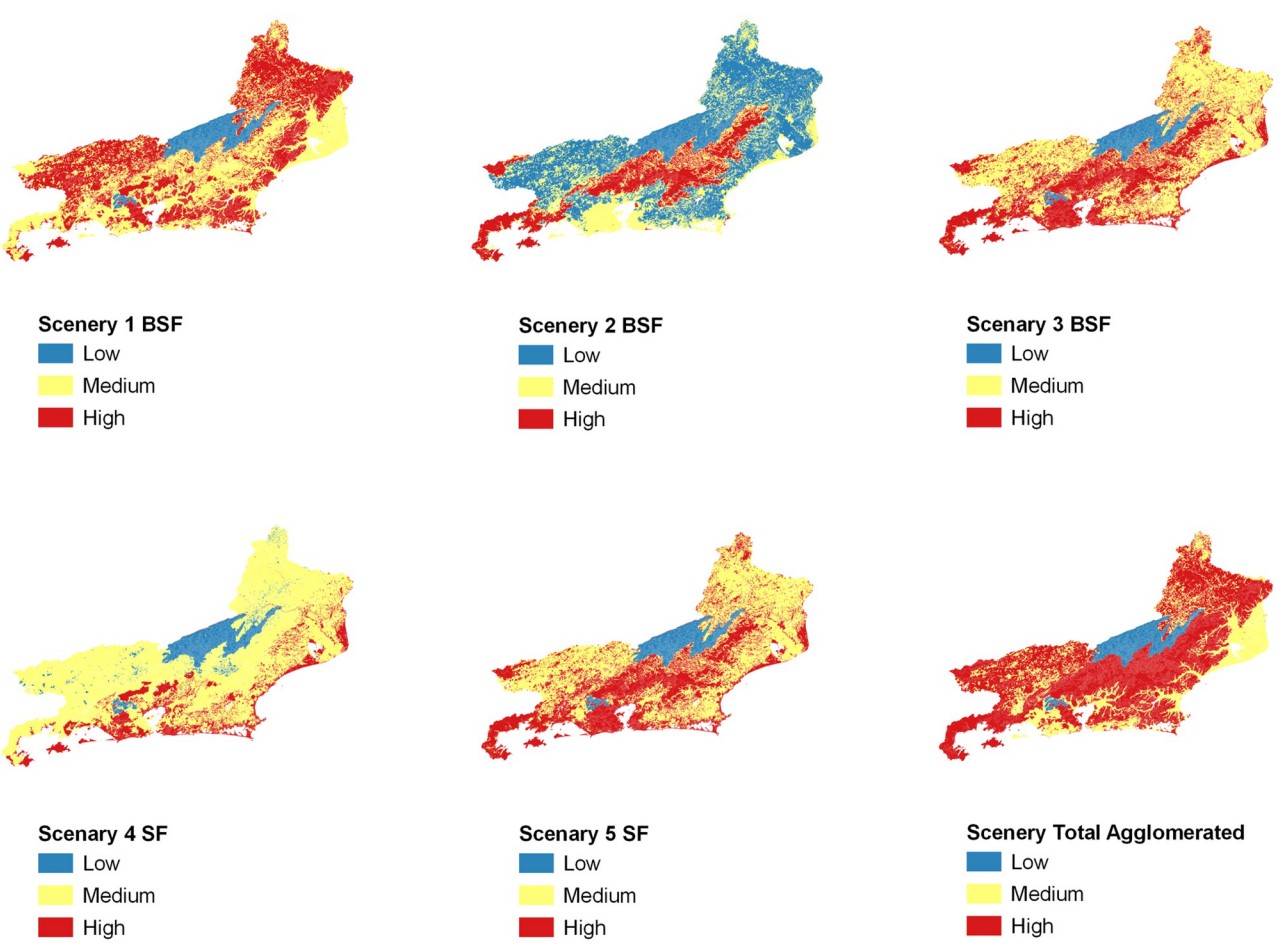

**Fig 3. Spotted fever rickettsiosis potential risk map to Rio de Janeiro state, Brazil due to ecological relationships between hosts and vectors.**
Scenery 1 (BSF); Scenery 2 (BSF); Scenery 3 (BSF); Scenery 4 (SF); Scenery 5 (SF); Scenery Total Agglomerated; Three colours' scale: blue colours—low potential risk; yellow colours—medium potential risk; red colours—high potential risk.

identification and quantification of specific immunoglobulins of the IgM and IgG class, which increase in titer with the evolution of the disease. The result should always be interpreted based on the clinical and epidemiological context related to the suspected case, since IgM antibodies can cross-react with other diseases (*e.g.*, dengue and leptospirosis) and, in general, perform better between the seventh and tenth day of illness. In addition to this method, it is possible to carry out the diagnosis by direct research using the rickettsial agent, in the case of Histopathology or Immunohistochemistry, performed on tissue samples obtained from biopsy of skin lesions of infected patients, especially those with severe symptoms or on tissue necropsy. Other way is the molecular biology technique with polymerase chain reaction (PCR), so allows a better and more adequate characterization of the two groups of rickettsiae: the Spotted Fever Group, to which *Rickettsia rickettsii*, *Rickettsia parkeri*, *Rickettsia africae*, *Rickettsia conorii* complex, among others belong; and the Typhus Group, constituted by *Rickettsia prowazekii* and *Rickettsia typhi* [2].

To validate the adopted methodology, we sought to use epidemiological data related to hospitalization and death indicators to translate severity factors related to rickettsiosis diseases. As it is not possible to continuously observe the circulation of etiological agents in the study area, it was decided to establish an epidemiological link between the severe cases, defined as those

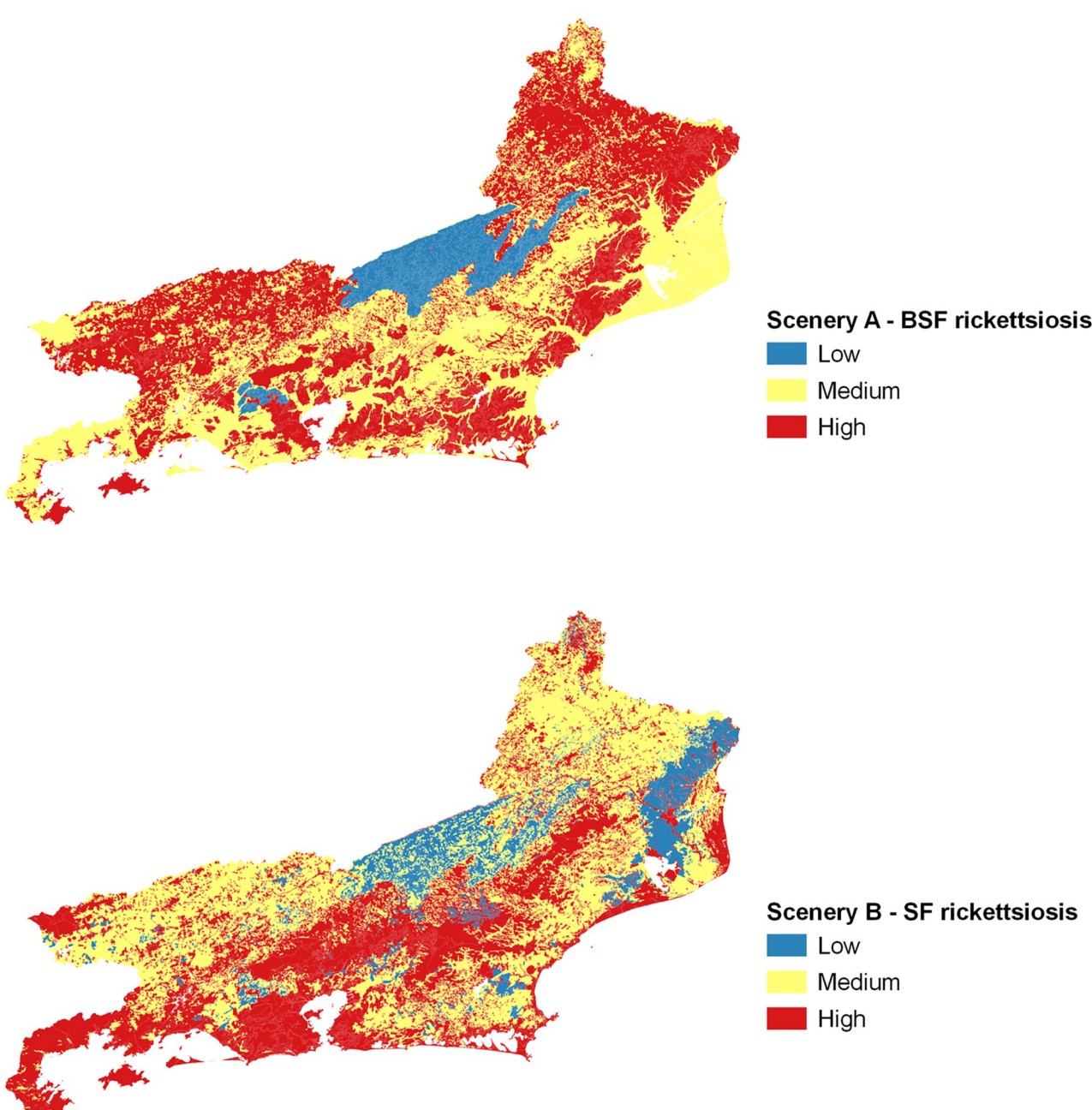

**Fig 4. Spotted fever rickettsiosis potential risk Map to Rio de Janeiro state, Brazil.** Synthesis Sceneries A (BSF) and B (SF) Scenery A (BSF); Scenery B (SF); Three colours' scale: blue colours—low potential risk; yellow colours—medium potential risk; red colours—high potential risk.

that required hospitalization and / or that caused death, with the presence of *Rickettsia rickett-sii*, recognized as the etiological agent of BSF rickettsiosis [10]. Mild or oligosymptomatic cases, but notified to the Information System for Notifiable Diseases (SINAN—Sistema de Informação de Agravos de Notificação) of the Brazilian Ministry of Health [30], were considered cases of SF rickettsiosis caused by others bacteria of the genus *Rickettsia* other than *Rickettsia rickettsii*.

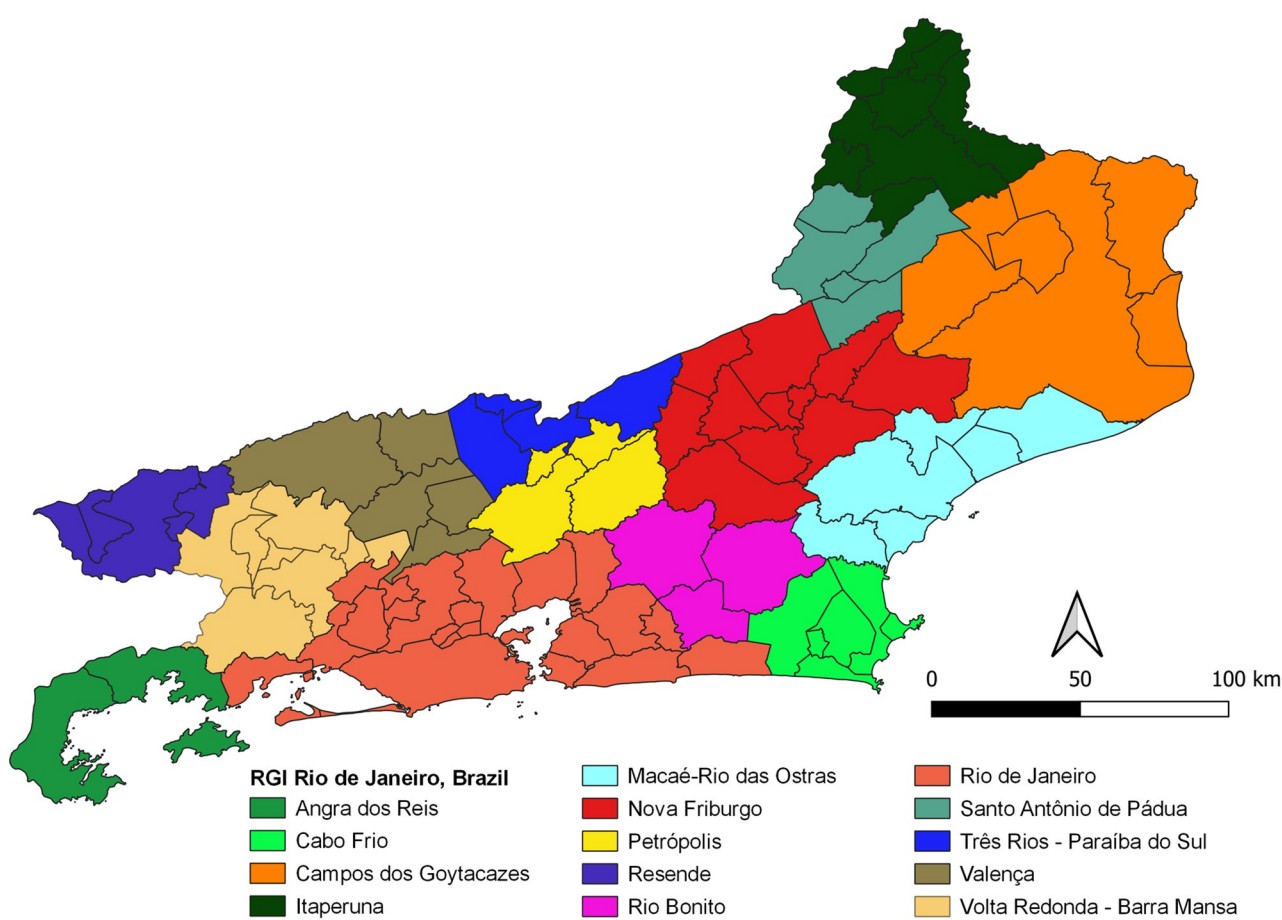

**Fig 5. Classification of Immediate Geographic Region (RGI) to Rio de Janeiro state, Brazil (IBGE).** Immediate Geographic Region, Rio de Janeiro state, Brazil and respective colours' scale. Adapted from an open access shapefile available at <https://inde.gov.br/AreaDownload#>, access Jan 28 2022 and Metadata ID: 515e67ee-ba6a-417b-bf53-511e5825d7cd.

A descriptive epidemiological study was carried out based on the completion of investigation forms for Spotted Fever rickettsiosis by the Local Health Surveillance Services, analyzing data regarding the geographic location (municipalities of notification of the occurrence of the case, of hospitalization of the patient and the probable site of infection) and those related to hospitalization, diagnosis, case outcome and clinical symptoms of patients. The software Tab for Windows (Tabwin®) version 4.15 and MS Excel® version 2010 were used for tabulation and statistical analysis of the data.

In order to validate the potential risk maps here produced, it was decided to plot notified cases, hospitalizations and deaths in an aggregated way in time and space using the most recent regional geographic classification, called Immediate Geographic Region (RGI—Região Geográfica Imediata), published by the official geography and statistics agency in Brazil (IBGE—Instituto Brasileiro de Geografia e Estatística) in 2017 [31] (Fig 5). Thus, we developed a spatial analysis process comparing the geographic layers of the epidemiological data obtained to the maps of potential risk for BSF and SF rickettsiosis for the immediate geographic regions of the Rio de Janeiro state.

### Ethical statement

Panelists who participated in the study have expressly agreed to offer data for research purposes, including the prospect of publishing results in a scientific article format. As it is an evaluation related to the surveillance of spotted fever carried out from the analysis of secondary data contained in a national and non-nominal database, the study was exempted from submission to a Research Ethics Committee (CONEP/CEP), supported by the texts of two Resolutions of the National Health Council: CNS nº 466, of December 12, 2012 (https://conselho.saude.gov.br/resolucoes/2012/466_english.pdf) and CNS nº 510, of April 7, 2016 (https://www.in.gov.br/materia/-/asset_publisher/Kujrw0TZC2Mb/content/id/22917581). The patient data that would allow them to be identified (patient's name, patient's mother's name, patient's residence address and respective zip code) were previously removed *ex officio* by the Ministry of Health of Brazil, through the Health Surveillance Secretariat (SVS), responsible for its custody. Access to SINAN-MS database used in the study was possible with institutional consent through request nº 25820.001767/2018-61, according to the Access to Information Law nº 12527, of November 12, 2011 (http://www.planalto.gov.br/CCIVIL_03/_Ato2011-2014/2011/Lei/L12527.htm).

## Results

In a Scenery BSF A (1, 2, 3) more hospitalizations and / or deaths were expected, depending on the greater circulation of the etiologic agent *Rickettsia rickettsii*. In its turn, in Scenery SF B (4, 5), an inverse behaviour was expected given the milder clinical characteristics of the rickettsiosis disease due to the fact that the etiologic agent involved in the transmission was not *Rickettsia rickettsii*. In this way, we chose to offer the results according to the list of scenarios that were presented above, being arranged in a sequence from 1 to 5.

Scenery 1 represents a potential risk for the transmission of Brazilian Spotted Fever rickettsiosis through ticks of the species *Amblyomma sculptum*. Due to the wide distribution in the study area and the recognized food promiscuity, we observed a great interaction with the selected hosts, which favoured a very large territory with a high potential risk for acquiring the disease in Rio de Janeiro state.

Scenery 2 corresponds to the potential risk for the transmission of Brazilian Spotted Fever rickettsiosis through ticks of the species *Amblyomma aureolatum*. It is interesting to note that there is almost no overlap with the areas of greatest potential risk offered in Scenery 1, similar to the negative of an exposed photograph, which can be explained by the characteristics of the preferred habitat of the species in question and its lower propensity to repast blood, reducing the number of host species. The mountain region and the southern coast of Rio de Janeiro state are the areas with the greatest potential risk for disease transmission.

Scenery 3 is responsible for the potential risk for the transmission of Brazilian Spotted Fever rickettsiosis through ticks of the species *Rhipicephalus sanguineus*, which have the domestic dog as their preferred host, which can be translated by a high potential risk for the transmission of the disease in areas of population agglomerations of Rio de Janeiro state, since, in general, domestic dogs have some dependence on humans.

Scenery 4 represents a potential risk for the transmission of Spotted Fever rickettsiosis through ticks of the species *Amblyomma ovale*. Its distribution in the study area is limited to lower altitudes, which, associated with the fragile interaction with selected hosts, with the exception of the domestic dog, translates into a discontinuous and small territorial strip, which encompasses the coast and the metropolitan region of Rio de Janeiro state, with high potential risk for acquiring the disease.

Scenery 5 represents a potential risk for the transmission of Spotted Fever rickettsiosis through ticks of the species *Rhipicephalus sanguineus*. Its distribution in the study area is limited to areas related to human agglomerates, since the domestic dog is its preferred host. In this way, we have a result with total similarity to what was achieved in Scenery 3, since the ecological relationships are the same, only varying the etiological agent that causes rickettsial disease.

Finally, the scenery AT reflects a high potential risk for the transmission of rickettsiosis through the four tick species evaluated in the territory of Rio de Janeiro state. Without taking into account the presence of the etiological agent, we observed that the study area has a high propensity for the transmission of the rickettsiosis in question, since the ecological relationships between vectors and hosts are, to a certain extent, preserved.

The scenery results that mention the potential risk for BSF rickettsiosis are strongly associated with representative RGIs in the Center-South and Northwest of Rio de Janeiro state. The results referring to the scenario representing the potential risk for SF rickettsiosis represent a large area of the coastal strip, especially Costa Verde, in the Southwest (RGI Angra dos Reis) and Metropolitan (RGI Rio de Janeiro) regions. The area comprising the RGIs of Paraíba do Sul—Três Rios and Nova Friburgo, on the border with Minas Gerais state (MG), corresponds to the low-risk potential area for BSF rickettsiosis. With some parsimony, the same can be observed for a considerable area of the RGI of Campos dos Goytacazes, located further to the North of Rio de Janeiro state, characterizing a low-risk potential for SF rickettsiosis.

Describing the results achieved based on geoenvironmental domains, directly related to the presence of important ticks for the maintenance of the life cycle of rickettsiae and closely related to BSF and SF in Brazil, we intuitively suggest that could be evaluated as a potential risk or as a protective factor for rickettsial diseases, as shown in Fig 6. Although it still needs further details in future studies, we observed that plateau and escarpment regions of Serra da Mantiqueira Mountain (RGI Resende) and Norte Fluminense (RGI Itaperuna), concomitantly, seem to act as protective factor for BSF rickettsiosis and as a potential risk factor for SF rickettsiosis. The plateau and escarpment regions of Serra dos Órgãos Mountain (RGIs Valença, Petrópolis and Nova Friburgo) and Serra da Bocaina Mountain (RGI Angra dos Reis) apparently function as a potential risk factor for rickettsial FS, also observed in the rift region of Guanabara (RGI Rio de Janeiro). Finally, the interplanaltine depressions of the Medium Paraíba do Sul and Pomba-Muriaé river basins would function as an important protective factor for FS rickettsiosis.

Between 2007 and 2016, 122 cases of spotted fever rickettsiosis in Rio de Janeiro state were reported to SINAN-MS, are related to Rio de Janeiro city and the central-south and northwest regions of the state, with 109 (89%) of hospitalizations and 46 (38%) of deaths. Of this universe, 80 patients (66%) were diagnosed by laboratory criteria, and for the others clinical-epidemiological criteria were used to confirm the occurrence. One hundred and six patients (87%) reported having had contact with ticks in the last 14 days before the symptoms.

We organize an illustration with three cartograms that relate epidemiological parameters (occurrence of cases, hospitalizations and deaths) arranged in the study territory in a weighted way (Low, Medium and High), facilitating the process of understanding and analyzing the situation for diseases tick-borne rickettsiae in Rio de Janeiro state (Fig 7).

## Discussion

Assessing the situation of BSF in the Rio de Janeiro state, we observed that the areas where there is a greater potential risk to contract the rickettsiosis coincide with the RGIs that concentrated the cases of hospitalization and deaths from the disease, especially Itaperuna,

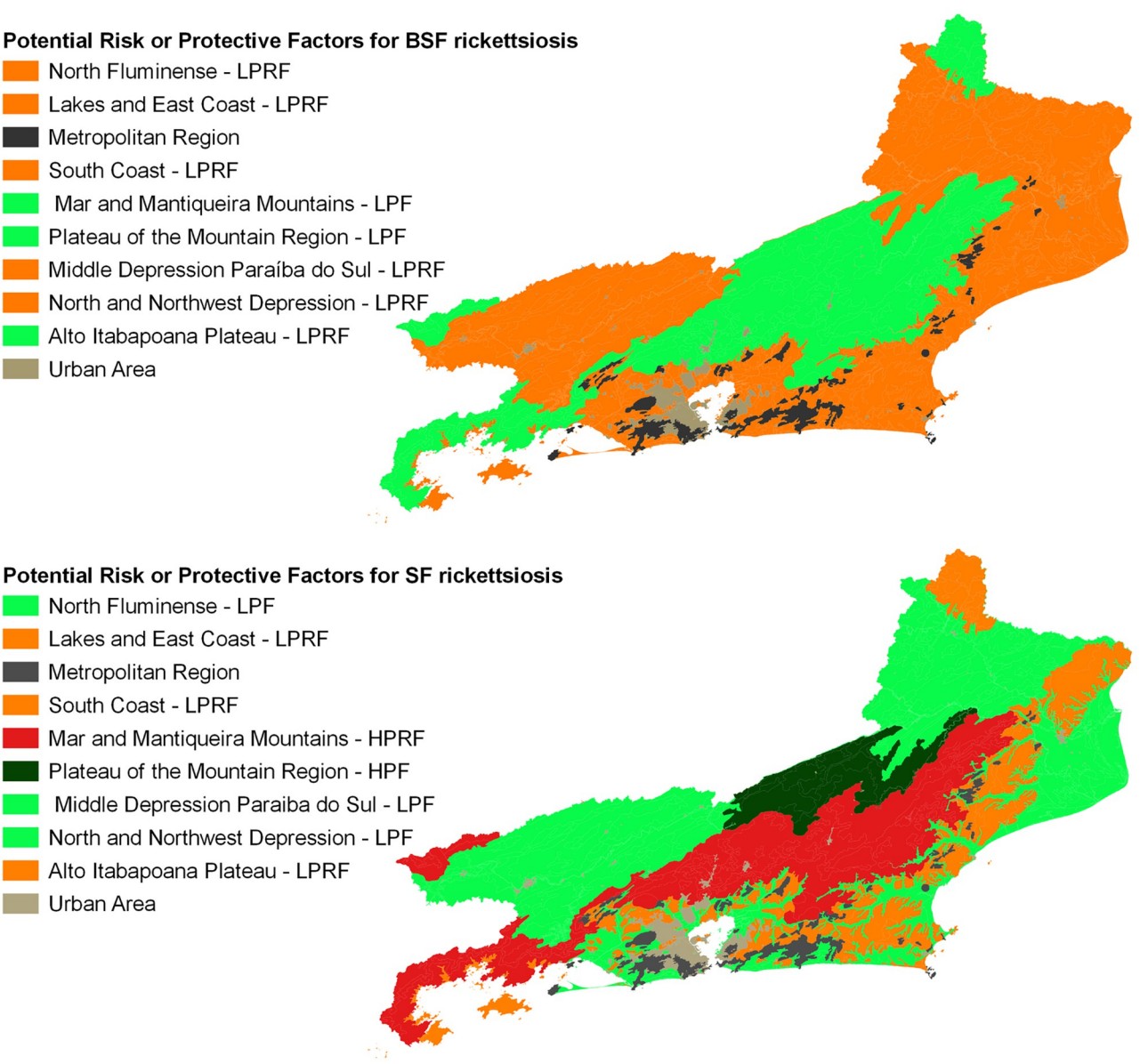

**Fig 6. Relation between geoenvironmental domain and potential risk or protective factor for spotted fever rickettsiosis, Rio de Janeiro state, Brazil.** BSF Rickettsiosis—Brazilian Spotted Fever (Scenery A), SF Rickettsiosis—Spotted Fever (Scenery B) with their respective colour scales (LPRF—Low Potential Risk Factor, LPF—Low Protective Factor, HPRF—High Potential Risk Factor and HPF—High Protective Factor).

Três Rios—Paraíba do Sul, Volta Redonda—Barra Mansa and Rio de Janeiro city. The exception is RGI Macaé—Rio das Ostras, which did not present any case notification during the studied period. The classification of the hazard factors for BSF in relation to the environmental domains evaluated corroborates the situation offered by the presented potential risk map, since the North-Northwest domains, the Medium Paraíba Valley and the Coastal Range overlap the aforementioned RGIs. Finally, of the ticks related to BSF transmission, we highlight the *Amblyomma sculptum*, which has wide distribution in Rio de Janeiro state [1, 9], enabling it to potentially cause rickettsial transmission in the aforementioned locations.

Although it is hypothetically possible for the transmission of BSF rickettsiosis by the vector *Amblyomma aureolatum* to occur in the study territory, no relationship between this species

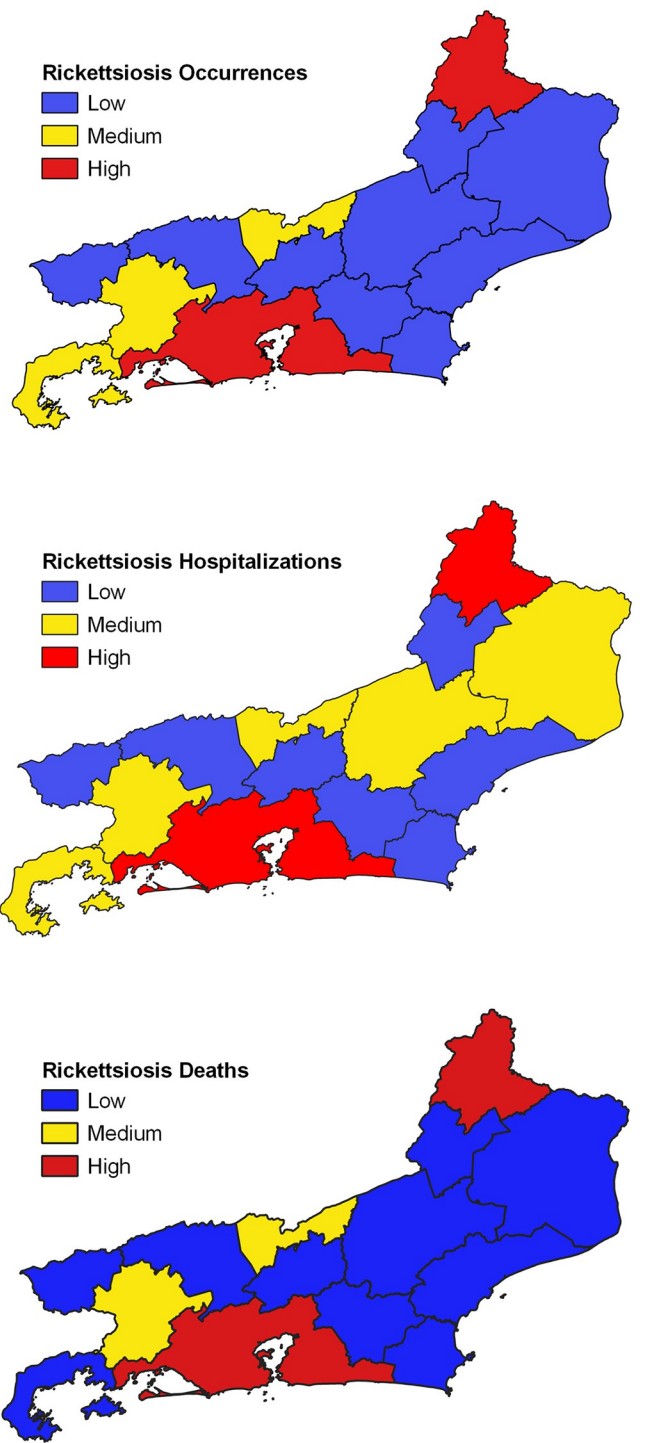

**Fig 7. Spotted fever occurrences, hospitalizations and deaths between 2007 and 2016 in Rio de Janeiro state, Brazil.** Occurrences Cases; Hospitalizations; Deaths and respective Three-colour scale: blue colours—low; yellow colours—medium; red colours—high.

and occurrences of the disease has been observed by scientists and professionals from the Health Surveillance of Rio de Janeiro state.

As for SF, we observed that because the symptoms are associated with mild, mostly oligo-symptomatic cases, hospitalization and / or death among those affected are rare events. In Rio de Janeiro state, of all reported cases, a little more than 10% did not require hospitalization, and 38% died, which seems to be a consequence of the small spread of other etiological agents than *Rickettsia rickettsii*. Thus, we believe that the protection factors for SF would be present in the areas related to the geoenvironmental domains that correspond to the RGIs Itaperuna, Santo Antônio de Pádua and Campos dos Goytacazes (North-Northwest), Valença, Volta Redonda—Barra Mansa and Resende (Medium Paraíba River Valley) and Santo Antônio de Pádua, Nova Friburgo and Três Rios—Paraíba do Sul (Plateau of the Mountainous Region).

Ticks associated with this disease, especially *Amblyomma ovale*, have a wide distribution throughout the studied territory, favouring their participation in cases of SF [1, 10, 11]. The geoenvironmental domain Plateau of the Mountain Region, which spatially corresponds, almost entirely, to RGI Nova Friburgo, presented a low potential risk in both sceneries. However, despite this classification, there were cases of hospitalization and deaths in the territory, but to a lesser extent than in regions classified as being of greater potential risk, which may be a consequence of the flow of patients between neighbouring municipalities seeking shelter in the local health system. This hypothesis needs to be further assimilated based on studies regarding the flows agreed between local health managers and those patient flows that originate in informality or spontaneity, usually observed in Brazil.

As for the brown dog tick (*Rhipicephalus sanguineus*), we believe that this species is a potential vector for rickettsiosis diseases, since it is linked to the maintenance of life cycles of rickettsial species of epidemiological importance in Brazil and in the world [5, 10, 29] and is an opportune vector for the domestic or peridomiciliary transmission of rickettsial agents to the human and animal population, since it has a gregarious social behaviour and is strongly influenced by the human presence in the territory, in addition to frequenting the borders of forests or agricultural areas with urbanized areas. However, despite the potential risk for transmission, we still do not have robust scientific findings that can affirm that the relationship between domestic dogs and these ticks is important in the maintenance of BSF or SF rickettsiosis in Rio de Janeiro state.

As for the hosts, we took as models for our study the domestic dog, horse and capybara, at the expense of other rodents, marsupials and other vertebrate animals that could also interact in the complex cycle of life of three falls of the genus *Amblyomma*. However, epidemiological data provided by the Brazilian Ministry of Health, a veritable compilation of investigation sheets corresponding to compulsory notifications of rickettsiosis, as well as authors recognized for their work on the ecoepidemiology of spotted fever rickettsiosis, indicate the importance of the three selected species. [11, 32, 33].

We believe that these hosts represent the most distinct ecological scenarios of the evaluated territory, as they permeate urban, wild and rural areas, in addition to their respective ecotones, being related to land cover and use in our model, as they are more adaptable to continuous anthropization phenomena. As for the vectors, four tick species identified as having the greatest potential for the transmission of BSF and SF rickettsiosis among the human population, have a more particular relationship with the geoenvironmental domains, since they are ecologically more related to abiotic factors: humidity, temperature, altitude and sunlight.

As for the validation of the methodology using official data on the occurrence of cases, hospitalizations and deaths from spotted fever rickettsiosis in the study area, we observed that digressions were necessary regarding the classification of the etiological agents in question. It

was observed that the laboratory diagnosis offered by the National Network of Epidemiological Surveillance Laboratories, despite having a high power of precision, maintains a standard conduct of classifying the genus (*Rickettsia spp.*) and the identification of the species involved in the occurrence of cases is not routine. It is also important to note that the sites of probable infection are still considered imprecise, despite all efforts by professionals from the local Environmental Surveillance Service to define them better.

Thus, we chose to make approximations in our analyses, supported by the clinical-epidemiological and environmental diagnoses performed by the local Health Surveillance teams, associating cases considered more serious, in which hospitalization and / or death of the patient occurred, as a proxy for *Rickettsia rickettsii* infection and, conversely, with no worsening of the clinical condition that would require hospitalization of the patient or that would allow a cure outcome, as a proxy for infection by another rickettsial species, as in the case of *Rickettsia parkeri*. We understand that the option for this methodological design is not a limitation that invalidates or weakens our study, on the contrary, it offers very creative epidemiological contours that indicate the need for more accurate biotic data with the innovation of local environmental and epidemiological surveillance strategies.

## Conclusion

Through spatial heuristics, we organize a binary model that links abiotic (land use and coverage and geoenvironmental domains) to biotic factors (vectors and hosts) and which, projected in the spatial or geographic plan, are associated with the ecological cycles that keep transmission active of spotted fever to humans. Thus, without being able to guarantee that the ticks are infected, as well as if there is a circulation of the etiological agents in the study area, we seek to produce maps of potential risk to measure the possibility of the occurrence of disease transmission. Recognizing that there are limitations in the adoption of methodology on a conceptual basis, we believe that it is an unprecedented and very important contribution to health management, since BSF is responsible for a historical series of deaths and hospitalizations, generating burdens for both families and for the brazilian health system.

Therefore, with the continuous and critical use of the here proposed tool, we expect an improvement of the techniques used, urging the more specific considerations of specialists and managers for new uses. We believe that the map of potential risk is a flexible model to the demands of the Health Services, with adoption by other units of the federation, for other diseases or for new abiotic variables, in addition to the possibility of expanding its complexity due to the inclusion of other dimensions of analysis, characterizing the multidimensionality of the model, which will allow the construction of sceneries that are increasingly consistent with the complexity of the reality of vector-borne diseases.

The analyses presented sought to subjectively determine how much each abiotic variable and its decomposition into domains or classes contributed to establish the potential risk for the transmission of spotted fever in Rio de Janeiro state. We observed that there are important distinctions between both sceneries (BSF and SF) in the same geographical space. It was also evaluated that the intersections of the input variables, when analysed in the logic of the proposed model, promote both potential risk and protection factors to contract rickettsial diseases. Finally, the validation of the methodology made use of epidemiological and ecological data available from official sources, bringing more reliability, reproducibility and flexibility potential to the model. Thus, we conclude that the sceneries found in Rio de Janeiro state are relevant to the case series for the disease and that the planning of surveillance actions can gain in quality if incorporated into the routine of local health management.

## Supporting information

**S1 Fig. Instrument for collecting expert data on biotic and abiotic variables related to the transmission of rickettsial diseases in Rio de Janeiro state, Brazil.**
(TIF)

**S1 Table. Balanced score according to experts for the relationship of vectors and hosts with rickettsial diseases in Rio de Janeiro state, Brazil.** AT—Aggregate Total = all weights the same, Scenery 1—only vector *Amblyomma sculptum* + dog + capybara + horse, Scenery 2—only vector *Amblyomma aureolatum* + dog, Scenery 3—only vector *Rhipicephalus sanguineus* + dog, Scenery 4—only vector *Amblyomma ovale* + dog + capybara + horse, Scenery 5—only vector *Rhipicephalus sanguineus* + dog.
(PDF)

**S2 Table. Final Score according to experts for the relationship of vectors and hosts with rickettsial diseases in Rio de Janeiro state, Brazil.** Legend: score vh = refers to individualized ecological relationship between vector and host, score vh total = refers to the specific relationship between vector and transmitted disease and scale of 100.
(PDF)

**S3 Table. Expert weights for geoenvironmental domains versus vectors in Rio de Janeiro state, Brazil.** Asculp—*Amblyomma sculptum*, Aaureo—*Amblyomma aureolatum*, Aovale—*Amblyomma ovale* and Rsang—*Rhipicephalus sanguineus.*
(PDF)

**S4 Table. Expert weights for coverage and land use versus hosts in Rio de Janeiro state, Brazil.**
(PDF)

**S1 File.**
(PDF)

**S2 File.**
(PDF)

**S3 File.**
(PDF)

**S4 File.**
(PDF)

**S5 File.**
(PDF)

## Acknowledgments

We thank MSc. Marcelo Eduardo Dantas, representing the other employees assigned to CPRM—Geological Service of Brazil, for offering databases and shapefiles that we use to organize the Geoenvironmental Domains layer used in the production of the presented potential risk maps and to panelists Adriano Pinter dos Santos, Alvaro A. Faccini-Martinez, Ana Beatriz Borsoi, André de Abreu Rangel Aguirre, Felipe da Silva Krawczak, Graziela Tolesano-Pascoli, Joana de Albuquerque Ribeiro, Jonas Moraes Filho, Karla Bitencourth Garcia, Marcelo Bahia Labruna, Matias Pablo Juan Szabó and Stefan Vilges de Oliveira, the latter also to validate the data collection instruments used in this manuscript.

## Author Contributions

**Conceptualization:** Cláudio Manuel Rodrigues, Francisco Dourado, Gilberto Salles Gazêta.

**Data curation:** Cláudio Manuel Rodrigues.

**Formal analysis:** Daniel Savignon Marinho.

**Investigation:** Cláudio Manuel Rodrigues.

**Methodology:** Francisco Dourado.

**Supervision:** Gilberto Salles Gazêta, Lena Geise.

**Validation:** Daniel Savignon Marinho.

**Visualization:** Cláudio Manuel Rodrigues, Francisco Dourado.

**Writing – original draft:** Cláudio Manuel Rodrigues.

**Writing – review & editing:** Francisco Dourado, Daniel Savignon Marinho, Gilberto Salles Gazêta, Lena Geise.

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
