## [Decision Letter · Decision Letter 0]

23 Nov 2021

PONE-D-21-18008Proposal for mapping potential risks for the transmission of spotted fever: the case study from the State of Rio de Janeiro, BrazilPLOS ONE

Dear Dr. RODRIGUES,

Thank you for submitting your manuscript to PLOS ONE. After careful consideration, we feel that it has merit but does not fully meet PLOS ONE’s publication criteria as it currently stands. Therefore, we invite you to submit a revised version of the manuscript that addresses the points raised during the review process.

The manuscript was revised by experts in the field and it requires major revision before can be considered for publication in the journal.  Please submit your revised manuscript by Jan 07 2022 11:59PM. If you will need more time than this to complete your revisions, please reply to this message or contact the journal office at plosone@plos.org. Please include the following items when submitting your revised manuscript:A rebuttal letter that responds to each point raised by the academic editor and reviewer(s). You should upload this letter as a separate file labeled 'Response to Reviewers'.A marked-up copy of your manuscript that highlights changes made to the original version. You should upload this as a separate file labeled 'Revised Manuscript with Track Changes'.An unmarked version of your revised paper without tracked changes. You should upload this as a separate file labeled 'Manuscript'.

We look forward to receiving your revised manuscript.

Kind regards,

Antonio Humberto Hamad Minervino, Ph.D.

Academic Editor

PLOS ONE

Additional Editor Comments (if provided):

The manuscript was revised by experts in the field and it requires major revision before can be considered for publication in the journal.

Journal Requirements:

2. In your ethics statement in the manuscript and in the online submission form, please provide additional information about the patient dataused in your study. Specifically, please ensure that you have discussed whether all data were fully anonymized before you accessed them.

4. We note that Figures 1, 2, 3, 4, 5, S1, and S2 in your submission contain [map/satellite] images which may be copyrighted. All PLOS content is published under the Creative Commons Attribution License (CC BY 4.0), which means that the manuscript, images, and Supporting Information files will be freely available online, and any third party is permitted to access, download, copy, distribute, and use these materials in any way, even commercially, with proper attribution. For these reasons, we cannot publish previously copyrighted maps or satellite images created using proprietary data, such as Google software (Google Maps, Street View, and Earth). For more information, see our copyright guidelines: http://journals.plos.org/plosone/s/licenses-and-copyright.

a. You may seek permission from the original copyright holder of Figures 1, 2, 3, 4, 5, S1, and S2 to publish the content specifically under the CC BY 4.0 license.  

Reviewers' comments:

Reviewer's Responses to Questions

**Comments to the Author**

1. Is the manuscript technically sound, and do the data support the conclusions?

Reviewer #1: Yes

Reviewer #2: Partly

2. Has the statistical analysis been performed appropriately and rigorously? 

Reviewer #1: I Don't Know

Reviewer #2: Yes

3. Have the authors made all data underlying the findings in their manuscript fully available?

Reviewer #1: Yes

Reviewer #2: Yes

4. Is the manuscript presented in an intelligible fashion and written in standard English?

Reviewer #1: No

Reviewer #2: Yes

5. Review Comments to the Author

Reviewer #1: Overall comment for the editor: This is an interesting exploration of the spatial and risk of rickettsioses in a region of Brazil. It is valuable since there are few studies on mapping hazard conducted in Brazil. However, there are some issues that need to be addressed before this paper is ready for publication. The main weakness of the paper is that the author did not do a very good job explaining the methods and they missed the opportunity to go deeper and use the land use data to understand whether the risk of rickettsiosis was associated to human-dominated landscapes. In order to improve the methods, I suggest adding subtitles.

Also, the fact authors grouped cases base on milder (R. parkeri) and severe cases (R. rickettsii) may lead to inaccurate conclusions and may affect the results. The data is still interesting, but the author needs to at least point this out as one of the main limitations of the study and how testing humans by serological tests rather than molecular approaches may led to cross reaction between Rickettsia species.

TITLE:

Remove “proposal”

Spotted fever group? Or spotted fever rickettsiosis? I suggest clarifying what the authors are referring to. I would suggest using the pathogen name rather the diseases since it is the pathogen the one that is actually transmitted.

ABSTRACT

Overall comments: The main results are missing and how these ones contribute to the system. Authors mentioned there were risk and protection factors for spotted fever, but they missed to mention which one they were.

Line 86: Spotted fever is actually a group rather than just one infectious disease. However, whether the authors are referring to Brazilian spotted fever transmitted by R. rickettsii or all spotted group rickettsioses need to be clarified.

Line 92: “risk” instead of “danger”

INTRODUCTION:

Overall comments: The authors listed several studies on hazard mapping, but it would be great if they can briefly explain what their main results are and how they contribute to the knowledge and to better understand the risk factors of pathogens transmission.

Line119-121: I suggest clarifying which ticks are the main vectors of R. parkeri and R. rickettsii.

Line 121: “genus Amblyomma” instead of “Amblyomma gender”

Line 126-127: The statement is confusing. Please clarify

MATERIAL AND METHODS:

Lines 145-150: It is not clear where the data come from and how the authors got the data of vectors and hosts. Also, were the tick and host data associated with Rickettsia spp. presence? It is important that the authors explain how human rickettsiosis cases were detected (e.g. molecular, serological).

Line 184-186: It seems that the authors use the presence of certain tick and host species, but they do not explain why they choose those ones and based on what the values were assigned to each one.

Line 186: What values 1 and 5 stand for? Please clarify

Line 211-213 (Table 2): The table was written in Portuguese rather than in English.

Line 225: Please explain what the nosological classification of the disease was ?

RESULTS:

It seems the authors missed to explain all the results such as the figures 2 and 3, also it seems the figure 4 should be before figure 2 and 3 since describe the study area.

Discussion:

Line 358: rickettsiosis instead riquetsiasis.

Line 376-377: host and Rickettsioses

Line376-380: It is hard to follow.

Reviewer #2: The results found in the article need to be described in more detail, the same occurring with the discussion, in which the data discussed are not fully consistent, especially in the choice of vertebrate and possible amplified animals that were used as a variable for classification of the area of risk. It is also interesting to detail what led to the classification of the disease into two forms (BSF and SF) and the possible vectors that were also chosen as a variable for classifying the area as at risk. I also missed A. aureolatum in the discussion of results.

6. PLOS authors have the option to publish the peer review history of their article (what does this mean?). If published, this will include your full peer review and any attached files.

Reviewer #1: No

Reviewer #2: No

---

## [Author Response · Author response to Decision Letter 0]

29 Mar 2022

Dear editor and reviewers, 

Upon receiving the reviewers' suggestions, we observed how opportune the reformulation was, not only of a group of sentences, but of considerable parts of the manuscript in question, since we understand it as a (good) challenge to bring up the issues related to the organization of our scientific thinking not yet laid out in the text. We bet on the reorganization of the Materials and methods section in partitions that would facilitate the understanding of the step-by-step that we built to reach the results. We also chose to add more information to the Introduction and reformulate the Results and Discussion sections in order to offer readers different perspectives related to vector-host interactions demonstrated in each of the potential hazard maps produced. For a better understanding of how much we worked on the new writing of the manuscript, we offer below a list of responses to the reviewers' suggestions.

#Reviewer 1

In general:

1. The main weakness of the article is that the author did not do a very good job of explaining the methods

Answer: We rewrite the section in question to give more clarity to the method used in the study. We have separated the section into partitions that best express the methodological design of the study.

2. Loss of opportunity to deepen and use land use data to understand the risk of rickettsiosis being associated with human-dominated landscapes.

Answer: By rewriting the study we tried to give the reader a better understanding of the relationships between vectors and hosts with the geographic layers chosen to produce the maps of potential risk.

3. To improve methods, add subheadings.

Answer: Suggestion accepted. We made uses of subheadings in the section.

4. Grouping cases based on milder (Rickettsia parkeri) and more severe (Rickettsia rickettsii) symptoms can lead to inaccurate conclusions and affect results. The data is still interesting, but it is necessary to at least point this out as one of the main limitations of the study and how testing humans by serological tests, rather than using molecular approaches, can lead to a cross-reaction between Rickettsia species.

Answer: In this revised version of the manuscript, we seek to offer a broader view of the distinction between rickettsial species involved in spotted fever transmission in Rio de Janeiro state. We understand that the use of epidemiological data is essential to distinguish between possible etiological agents, since the official laboratories of the Brazilian national epidemiological surveillance network do not distinguish between the species involved, only focusing on the diagnosis of the genus of the bacterium that causes the disease. It was observed that the gold standard for diagnosing rickettsiosis is the indirect immunofluorescence test, which, despite having high specificity, can actually lead to cross-reactions between rickettsiae of the same group (heterotypic antibodies), as is the case with Rickettsia rickettsii and Rickettsia parkeri.

TITLE

1. Remove “proposal”

Answer: Suggestion accepted.

2. Spotted fever group? Or spotted fever rickettsiosis? I suggest clarifying what they are referring to. Suggestion: use the name of the pathogen instead of the diseases, as it is the pathogen that is actually transmitted.

Answer: Yes, the idea is to work with spotted fever rickettsiosis. Alerted to this possible conceptual confusion, we chose, throughout the text, to clearly express our intention. We work with the pathogens, but also with the diseases linked to them (Rickettsia rickettsii -> BSF rickettsiosis and Rickettsia parkeri -> SF rickettsiosis).

ABSTRACT

1. General comments: Key results and how they contribute to the system are missing. The authors mentioned that there were risk and protective factors for spotted fever rickettsiosis, but failed to mention what they were.

Answer: We follow the suggestion

2. Line 86: Spotted fever rickettsiosis is actually a group of diseases and not just an infectious disease. However, whether the authors are referring to Brazilian spotted fever transmitted by R. rickettsii or to all rickettsiosis of the spotted fever group needs to be clarified.

Answer: We follow the suggestion.

3. Line 92: “risk” rather than “danger”

Answer: Although we consider different concepts terminologically, we follow the suggestion.

INTRODUCTION

In general: the authors have listed several studies on hazard mapping, but it would be great if they could briefly explain what their main findings are and how they contribute to knowledge and better understanding of risk factors for pathogen transmission.

Answer: we added a paragraph to the text referring specifically to the three previously cited articles.

1. Line 119-121: Suggestion: clarify which ticks are the main vectors of R. parkeri and R. rickettsii.

 Answer: Suggestion accepted.

2. Line 121: "genus" instead of "genre"

Answer: Adjusted. 

3. Line 126-127: The instruction is confusing. Please clarify.

Answer: The text was rewritten in order to give more clarity to the message.

MATERIAL AND METHODS

1. Line 145-150: It's not clear where the data comes from and how the authors got the data from vectors and hosts. In addition, data on ticks and hosts associated with Rickettsia spp. presence? It is important that authors explain how human rickettsiosis cases were detected (e.g. molecular, serological).

Answer: We sought to rewrite the text to clarify the origin of the data used in the analysis related to vectors, hosts and rickettsiosis occurrences in humans. It is not possible to obtain data on rickettsial infection in ticks related to all suspected cases in humans, as these vectors are not collected by local environmental health surveillance services other than after laboratory diagnosis or through deduction by the medical clinic and association with the probable site of infection by the local epidemiological surveillance service. That is why we make use of the approximation of the more or less severe clinical presentation with the possible etiologic agent involved. At one point in the text (Epidemiological Validation) we have the information that the gold test for the disease in Brazil is the indirect immunofluorescence reaction of paired samples (serological), but it is also possible to perform molecular analysis from the Polymerase Chain Reaction (PCR).

2. Line 184-186: It seems that the authors use the presence of certain tick species and hosts, but do not explain why they chose these species and on the basis of what values were assigned to each.

Answer: The entire section was rewritten, reorganized in its presentation, to clarify the methodological design adopted and to provide robustness to the results found. In addition, as a supplementary material, the set of spreadsheets that offers the calculation and operationalization mechanisms of the adopted weighting models was offered.

3. Line 186: What values 1 and 5 represent? Please clarify

Answer: As already mentioned, the section was reorganized to provide clarity on the methodological design adopted and to provide robustness to the results found.

4. Line 211-213 (Table 2): The table was written in Portuguese and not in English.

Answer: Adjusted.

5. Line 225: Please explain what was the nosological classification of the disease?

Answer: The word "nosological" was used arbitrarily in order to "characterize" or "classify" the disease, but it does not seem to be the best way to express our thoughts. Therefore, it was replaced to give more clarity to the text, using the expression "symptom severity", which, together with the new wording of the section, seemed more appropriate.

RESULTS

1. It seems that the authors failed to explain all the results, like figures 2 and 3, it also seems that figure 4 should be before figures 2 and 3, as they describe the study area.

Answer: As already said, by reorganizing the section, we believe that we have solved comprehension problems such as this statement. Some figures have been replaced, others redone or included in the course of the manuscript to give a better understanding of what we propose and to give fluidity to the reading.

DISCUSSION

1. Line 358: use “rickettsiosis”

 Answer: Adjusted. 

Line 376-377: host and Rickettsiosis

Answer: Adjusted. 

2. Line 376-380: It's hard to follow.

Answer: The text was rewritten in order to give more clarity to the message.

Reviewer #2:

1. The results found in the article need to be described in more detail, as well as the discussion, in which the data discussed are not fully consistent, especially in the choice of vertebrates and possible amplified animals that were used as a variable to classify the risk area. 

Answer: We accepted the criticism in a positive way, proposing a complete restructuring of the Results section, describing them in greater detail, taking care not to be too boring to give fluidity to the reading. The Discussion section was added with paragraphs that, in addition to seeking to give more emphasis to the results found, tried to achieve the desired consistency, including the choice of the four vectors and the three representatives of hosts used as a variable to classify the area of potential risk for rickettsiosis.

2. It is also interesting to detail what led to the classification of the disease in two forms (BSF and SF) and the possible vectors that were also chosen as a variable to classify the area as at risk. 

Answer: We rewrote the manuscript in order to offer an adequate explanation of the eco-epidemiological criteria that guide our methodological approach and that led us to classify the disease in two forms (BSF and SF), in addition to expressing more clearly the perspective of choosing the four tick species and three vertebrate hosts used as a variable to classify the area as a potential risk for rickettsiosis. We hope we got it right.

3. Amblyomma aureolatum is missing from the discussion of the results.

Answer: Adjusted. 

Best regards,

The authors

---

## [Decision Letter · Decision Letter 1]

27 Apr 2022

PONE-D-21-18008R1Mapping potential risks for the transmission of spotted fever rickettsiosis: the case study from the Rio de Janeiro state, Brazil.PLOS ONE

Dear Dr. RODRIGUES,

Thank you for submitting your manuscript to PLOS ONE. After careful consideration, we feel that it has merit but does not fully meet PLOS ONE’s publication criteria as it currently stands. Therefore, we invite you to submit a revised version of the manuscript that addresses the points raised during the review process.

Dear authors, the manuscript was revised by experts in the field and, although it was improved, there are still issues that need to be addressed. The manuscript can be considered for publication after a minor revision. The main issue was the new concern raised by reviewer#1 that we missed in the first revision. Please address all issues and provide a revised manuscript   Please submit your revised manuscript by Jun 11 2022 11:59PM. If you will need more time than this to complete your revisions, please reply to this message or contact the journal office at plosone@plos.org. Please include the following items when submitting your revised manuscript:A rebuttal letter that responds to each point raised by the academic editor and reviewer(s). You should upload this letter as a separate file labeled 'Response to Reviewers'.A marked-up copy of your manuscript that highlights changes made to the original version. You should upload this as a separate file labeled 'Revised Manuscript with Track Changes'.An unmarked version of your revised paper without tracked changes. You should upload this as a separate file labeled 'Manuscript'.If applicable, we recommend that you deposit your laboratory protocols in protocols.io to enhance the reproducibility of your results. Protocols.io assigns your protocol its own identifier (DOI) so that it can be cited independently in the future. For instructions see: https://journals.plos.org/plosone/s/submission-guidelines#loc-laboratory-protocols. Additionally, PLOS ONE offers an option for publishing peer-reviewed Lab Protocol articles, which describe protocols hosted on protocols.io. Read more information on sharing protocols at https://plos.org/protocols?utm_medium=editorial-email&utm_source=authorletters&utm_campaign=protocols.

We look forward to receiving your revised manuscript.

Kind regards,

Antonio Humberto Hamad Minervino, Ph.D.

Academic Editor

PLOS ONE

Journal Requirements:

Additional Editor Comments (if provided):

Dear authors, the manuscript was revised by experts in the field and, although it was improved, there are still issues that need to be addressed. The manuscript can be considered for publication after a minor revision. The main issue was the new concern raised by reviewer#1 that we missed in the first revision. Please address all issues and provide a revised manuscript

Reviewers' comments:

Reviewer's Responses to Questions

**Comments to the Author**

1. If the authors have adequately addressed your comments raised in a previous round of review and you feel that this manuscript is now acceptable for publication, you may indicate that here to bypass the “Comments to the Author” section, enter your conflict of interest statement in the “Confidential to Editor” section, and submit your "Accept" recommendation.

Reviewer #1: (No Response)

Reviewer #2: (No Response)

2. Is the manuscript technically sound, and do the data support the conclusions?

Reviewer #1: Partly

Reviewer #2: Yes

3. Has the statistical analysis been performed appropriately and rigorously? 

Reviewer #1: I Don't Know

Reviewer #2: Yes

4. Have the authors made all data underlying the findings in their manuscript fully available?

Reviewer #1: Yes

Reviewer #2: Yes

5. Is the manuscript presented in an intelligible fashion and written in standard English?

Reviewer #1: Yes

Reviewer #2: Yes

6. Review Comments to the Author

Reviewer #1: The authors improved the manuscript. However, there are still some issues that need to be fixed.

My main concern that I did not think about on my first review, is that I assumed the experts knew where the cases were reported, so when they gave the score values to the cover and land use and vector-host association it was likely that the spatial risk model would match where the cases are, potentially biasing the model results. I think it would be more valuable if the authors can discuss more about whether the risk maps are associated with the distribution or density of vectors and/or hosts, rather than the human cases.

ABSTRACT

Line 94: Please describe which ones were the risk and protection factors. Based on what the authors studied it seems the statement is not accurate, since the authors modeled only the risk spatially based on scores previously given to the association variables, but they do not aim to study the influence of those factors in the cycle.

Line 99: Sceneries is incorrect

INTRODUCTION:

Line 111: Spotted fever rickettsiosis is caused by a group of species

Line 115: humans instead of huma beings.

Line 122-127: seems that the explanation of choosing ticks would fit better in the methods section.

METHODS:

The methods are still a little bit hard to follow. For example, the authors did not mention how many types of landscape/habitats are within the study in the manuscript. They only mentioned them in the table 2. Same happens with geoenvironmental domains. I think the authors should briefly explain what they are in the methodological proposal section.

Line 276: I would suggest adding subtitles to the Expert panels section. It is hard to follow all weighting procedures.

Line 288: I wonder if there was any way to confirm the values giving by the experts base on reviewing the literature. This question is also related to the fact that capybara scores showed to be more likely in urban areas than rural areas. Does this make sense? If so, could you please explain it why?

Line 332: T is missing

Line 346: Please clarify what the peculiar characteristics are.

Line 454: Please add a reference for sensitivity and specificity. High specificity means it can differentiate between R. parkeri and R. rickettsii?

Line 474: Is there any reference that support the fact that R. rickettsii is not associated with any mild cases? If so, please add it.

DISCUSSION

Line 649: typo

Line 648: It is well known Rickettsia parkeri tend to cause mild cases and R. rickettsii more severe cases. However, I think it would be valuable to discuss about whether R. rickettsii can also be associated with mild cases as well. Assuming that all mild cases are caused by Rickettsia species other than R. rickettsii it may inaccurate.

Reviewer #2: After a new evaluation, I recommend the publication because the corrections, suggestions and changes were met by the authors.

7. PLOS authors have the option to publish the peer review history of their article (what does this mean?). If published, this will include your full peer review and any attached files.

Reviewer #1: No

Reviewer #2: No

---

## [Author Response · Author response to Decision Letter 1]

24 May 2022

Response Letter to Reviewers

Reviewer #1: The authors improved the manuscript. However, there are still some issues that need to be fixed.

a. We thank you for your kind observation. We were challenged to improve our academic writing to seek to clarify to the reader the motivations of our study, whether qualifying the introduction, reorganizing the methodology into legends or developing images to give a better understanding of our results.

My main concern that I did not think about on my first review, is that I assumed the experts knew where the cases were reported, so when they gave the score values to the cover and land use and vector-host association it was likely that the spatial risk model would match where the cases are, potentially biasing the model results. I think it would be more valuable if the authors can discuss more about whether the risk maps are associated with the distribution or density of vectors and/or hosts, rather than the human cases.

b. The panel of experts was designed from the perspective of:

1. All participants are experts in different areas of knowledge, which, when associated, could offer a more reliable result of the relationship between vectors and hosts with components related to soil cover and the geoenvironmental aspects of the study territory;

2. we expected the experts to analyze the ecological relationship between the animals that participate in the ecological cycle associated with rickettsiosis, that is, suspected or confirmed cases of the disease would not be observed in the procedures related to the expert panels; and

3. we did not control for this hypothetical bias, in which experts would be influenced by recognizing the points (exact or not) of infection (human cases), since, in fact, we will hardly have in the disease investigation sheets the exact location of contact with infected ticks (this yes, a problem to be solved by the management of health surveillance) and, certainly, very few (or none?) of these specialists would have access to investigation files because they are reserved for epidemiological and environmental surveillance of Brazilian institutional services.

c. The model we proposed is related to the spatial availability of the host / vector distribution. This is clear in the first paragraph to Methodological proposal: 

"Understanding that the potential risk for the transmission of spotted fever rickettsiosis (PRSF) can be demonstrated by the relationship between vectors (v) and hosts (h) projected in the geographical space, we formulated a methodological proposal".

ABSTRACT

Line 94: Please describe which ones were the risk and protection factors. Based on what the authors studied it seems the statement is not accurate, since the authors modeled only the risk spatially based on scores previously given to the association variables, but they do not aim to study the influence of those factors in the cycle.

d. We offer an editing of the manuscript text for better understanding of the reader: 

“The results achieved were substantially encouraging, considering that there are territories with greater or lesser expectation of risk for spotted fever in the study area”.

Line 99: Sceneries is incorrect

e. Corrected in the text (“scenery”)

INTRODUCTION

Line 111: Spotted fever rickettsiosis is caused by a group of species

f. We appreciate the suggestion, but we believe that the text is consistent with what we communicate to the reader. In the immediate sequence of the text, we identified that there is a group of vector species that transmit bacteria of the genus Rickettsia, etiological agents of Spotted Fever (SF) and Brazilian Spotted Fever (BSF).

Line 115: humans instead of human beings.

g. Corrected in the text (“human beings”)

Line 122-127: seems that the explanation of choosing ticks would fit better in the methods section.

h. Corrected in the text (“Considering this we selected Three tick species of the…”)

METHODS:

The methods are still a little bit hard to follow. For example, the authors did not mention how many types of landscape/habitats are within the study in the manuscript. They only mentioned them in the table 2. Same happens with geoenvironmental domains. I think the authors should briefly explain what they are in the methodological proposal section.

i. We emphasize that the characteristics related to land use and geoenvironmental domains are displayed in Figure 2 which, coupled with the legend and associated with the image in Figure 1, offers a view of both profiles in the study territory. In addition, we leave the access links to more information about it and the reference literature available in the manuscript text.

We understand that the reviewer's observation must be taken into account and, therefore, we offer an editing of the manuscript text for better understanding of the reader: 

“By definition, geoenvironmental domains represent morphostructures that relate to remarkable events, which are responsible for the current arrangement of the relief and for the less mutable characteristics of the landscape. From the perspective of our study, the geoenvironmental domains refer to a larger taxon, compatible with regions with the same geological and geomorphological characteristics that group the same habitats and their respective faunal and floristic communities [19]. 

 The geoenvironmental data used in this work are the result (…)”

Line 276: I would suggest adding subtitles to the Expert panels section. It is hard to follow all weighting procedures.

j. At the end of the first paragraph of the Expert Panel subtitle, we made a small edit in order to give more elegance to the text, but we understanding that two expert panels were held at different times, both portrayed in the Methodology Section. The first panel [line 306: "This primary weighting (...)"], which has the result portrayed in Table 1, used 16 panelists from different areas of knowledge and with experience related to rickettsial diseases that we discussed during the study. The methodological perspective is well described, paragraph after paragraph. The second panel [line 347: "In a second moment, four specialists(...)"] had 4 experts and developing the methodological procedures laid out in the text, reached the results expressed in Tables 2 and 3. Both panels are documented in supplemental material provided by the authors. Therefore, we consider that the text is clear and the use of subheadings would not add better understanding for the reader.

Line 288: I wonder if there was any way to confirm the values giving by the experts base on reviewing the literature. 

k. The choice to consult the specialists stems from the fact that there is a scarce literature focused on the ecological theme of the enzootic and epizootic cycles of rickettsiosis in the study area. Of the few epidemiological studies published in the last five years and related to the study area, we highlight the themes of altitudinal distribution of tick species of importance for the transmission of SF and BSF (https://doi.org/10.1186/s13071-022-05250 -6); factors related to animal hosts of certain tick species on Ilha Grande (https://doi.org/10.1080/01647954.2020.1804999); and eco-epidemiological scenario of spotted fever (https://doi.org/10.3389/fmicb.2017.00505). Thus, the methodological profile was structured in such a way as to make scientific arguments for use in the study based on the opinion of experts, who accumulate knowledge during their academic careers.

This question is also related to the fact that capybara scores showed to be more likely in urban areas than rural areas. Does this make sense? If so, could you please explain it why?

l. Like horses and domestic dogs, capybaras deserved to receive higher scores among the specialists consulted because, in the case of Rio de Janeiro state, they are present in urbanized areas of territory. It is noted that Rio de Janeiro state has a historical process of urbanization, since the time of Colonial-Brazil, it had economic, cultural and political importance, becoming the capital of the country for many centuries until, in the 1960s, the inauguration of Brasília, DF took place.

Dogs play an important role in the social life of humans, having been with us since they approached and were domesticated by the peoples of Southeast Asia over 30,000 years ago (www.sciencedaily.com/releases/2017/07/170718113516.htm). It is easy to understand that it is present in practically the entire territory studied, thus justifying the weight offered as an important host of ticks of epidemiological importance for rickettsial diseases in the study area.

Horses have a very similar historical perspective to dogs, justifying their presence in rural and peri-urban areas. Culturally, in areas of greater human density, it can be seen as a working animal, in the case of the mounted police, but also related to the low-income population, who live in precarious conditions and use their services for personal transport or household items, as well as scraps of constructions and other objects that can be transported on small metal and wooden boards, being an important source of income for this part of the population (http://dx.doi.org/10.4102/jsava.v91i0.2009).

Capybaras accompany the recent urbanization process not only of the study area, but of the entire country (https://www.bravietour.com.br/mangrove-and-jungle-tour ; https://doi.org/10.1016/j.landurbplan.2022.104398

 and https://link.springer.com/article/10.1007/s10493-012-9533-1). Specifically in Rio de Janeiro state, urban agglomerations take the place of old dairy farms or small peri-urban swiddens, which were once responsible for supplying vegetables and fruits to cities. Commonly associated with water collections (rivers, lagoons, etc.) and field areas, it is currently common for them to be found walking on the streets of cities in Rio de Janeiro state, including its capital, interacting with the urban landscape and, in a way, quickly adapting to new living conditions by finding food and security necessary to reproduce and generate offspring that guarantee the maintenance of future generations. It can be said currently, in addition to the ecological and environmental perspective, sanitary nuances have been a questioning factor for their presence in inhabited areas, since they are protected from slaughter by national legislation and because, in cities, they do not have natural predators (http://doi.org/10.1089/vbz.2019.2479 and https://doi.org/10.1093/jmammal/gyaa144).

Line 332: T is missing

m. Corrected in the text (“The”)

Line 346: Please clarify what the peculiar characteristics are.

n. We offer an editing of the manuscript text for better understanding of the reader: 

“(…) characteristics that differentiate them from each other (e.g., eating habits; degree of insolation, humidity and temperature; vegetation type; altitudinal range and anthropophilia), which (…)” 

and respective references [1, 10, 11].

Line 454: Please add a reference for sensitivity and specificity. High specificity means it can differentiate between R. parkeri and R. rickettsii?

o. Thanks for the inquiry. We edited the text for better understanding by the reader. We missed the opportunity to offer a more up-to-date reading of the procedures determined by the reference laboratories in the original text and took advantage of the description of the gold standard diagnosis for spotted fever by the brand-new edition of the Health Surveillance Guide (5th ed.) of 2021, which has just been published by the Ministry of Health, to elucidate the issue. We also updated the guide cited [2] in the References section.

The text was like this:

 “(…) This method comprises a high sensitivity and specificity reaction that can be used to identify and quantify specific immunoglobulins of the IgM and IgG class is established by the identification and quantification of specific immunoglobulins of the IgM and IgG class, which increase in titer with the evolution of the disease. The result should always be interpreted based on the clinical and epidemiological context related to the suspected case, since IgM antibodies can cross-react with other diseases (e.g., dengue and leptospirosis) and, in general, perform better between the seventh and tenth day of illness. In addition (…)”

Line 474: Is there any reference that support the fact that R. rickettsii is not associated with any mild cases? If so, please add it.

p. To our knowledge, there is no publication that categorically states that there is no possibility of non-severe cases of spotted fever caused by infection with the agent Rickettsia rickettsii. But we also don't know of any scientific publication that denies this possibility in an unquestionable way.

DISCUSSION

Line 649: Type

q. Editing: “(...) which seems to be a consequence of the small spread of other etiological agents than Rickettsia rickettsii.”

Line 648: It is well known Rickettsia parkeri tend to cause mild cases and R. rickettsii more severe cases. However, I think it would be valuable to discuss about whether R. rickettsii can also be associated with mild cases as well. Assuming that all mild cases are caused by Rickettsia species other than R. rickettsii it may inaccurate.

r. Yes, thanks for your question. Although it is very important for Health Surveillance to clinically recognize the disease, in the study we did not seek to discuss the clinical nuances linked to the different etiological agents that would cause rickettsiosis in the territory of Rio de Janeiro. We focus on producing knowledge about the life cycles of four tick species, and consequently, the rickettsiae related to them in a delimited geographic space, seeking to determine a possible spatial distribution of the potential danger in having contact with the disease. 

Currently, in Brazil, it is still a challenge to identify the species related to symptomatic cases of rickettsiosis, since, for purposes of epidemiological surveillance, it is enough to recognize the genus Rickettsia and confirm the suspicion notified by the local health management. However, the scientific literature shows that there is a group of rickettsiae that differs from the symptomatology and the medical clinic expected for cases related to Rickettsia rickettsii. These are oligosymptomatic cases, at best, cases of mild symptoms that progress very positively to cure, without major complications. In a way, the cases related to Rickettsia rickettsii and Rickettsia parkeri - Atlantic Forest strain are relatively well delimited spatially and also maintain a distinct correlation with the clinical symptoms presented by the patients.

In fact, there is no way to precisely determine whether all cases related to Rickettsia rickettsii are severe enough to lead patients to hospitalization and, sometimes, to death. However, it is recognized that with such distinct clinical characteristics, with such disparate symptoms, we would hardly have cases of FS (associated with Rickettsia parkeri) being confused with BSF (associated with Rickettsia rickettsii), even more recognized the eco-epidemiological scenery that we described in the study. The confusion could be due to the congruence of areas of occurrence of Amblyomma ovale, Amblyomma aureolatum and Amblyomma sculptum in Rio de Janeiro state and, thus, with the perspective of the circulation of different etiological agents in the territory. Although it is in a relatively small territory, but of high complexity regarding the abundance of natural landscapes and the types of relief demonstrated in the study by the geo-environmental profile and the cover and use of the soil, the study showed that it is possible to identify the areas affected by the types SF and BSF for the methodological proposal that we are offering.

To finish this answer, we explained the limitations of the study, including the aspect mentioned by the reviewer, in the text of the manuscript that we present below:

“As for the validation of the methodology using official data on the occurrence of cases, hospitalizations and deaths from spotted fever rickettsiosis in the study area, we observed that digressions were necessary regarding the classification of the etiological agents in question. It was observed that the laboratory diagnosis offered by the National Network of Epidemiological Surveillance Laboratories, despite having a high power of precision, maintains a standard conduct of classifying the genus (Rickettsia spp.) and the identification of the species involved in the occurrence of cases is not routine. It is also important to note that the sites of probable infection are still considered imprecise, despite all efforts by professionals from the local Environmental Surveillance Service to define them better”.

And, even more, in the last paragraph of the Discussion:

”Thus, we chose to make approximations in our analyses, supported by the clinical-epidemiological and environmental diagnoses performed by the local Health Surveillance teams, associating cases considered more serious, in which hospitalization and / or death of the patient occurred, as a proxy for Rickettsia rickettsii infection and, conversely, with no worsening of the clinical condition that would require hospitalization of the patient or that would allow a cure outcome, as a proxy for infection by another rickettsial species, as in the case of Rickettsia parkeri. We understand that the option for this methodological design is not a limitation that invalidates or weakens our study, on the contrary, it offers very creative epidemiological contours that indicate the need for more accurate biotic data with the innovation of local environmental and epidemiological surveillance strategies”.

Reviewer #2: After a new evaluation, I recommend the publication because the corrections, suggestions and changes were met by the authors.

s. Thank you for supporting us in this academic effort to disseminate our manuscript.

---

## [Decision Letter · Decision Letter 2]

20 Jun 2022

Mapping potential risks for the transmission of spotted fever rickettsiosis: the case study from the Rio de Janeiro state, Brazil.

PONE-D-21-18008R2

Dear Dr. RODRIGUES,

We’re pleased to inform you that your manuscript has been judged scientifically suitable for publication and will be formally accepted for publication once it meets all outstanding technical requirements.

Kind regards,

Antonio Humberto Hamad Minervino, Ph.D.

Academic Editor

PLOS ONE

Additional Editor Comments (optional):

Dear authors,

I am glad to inform that your manuscript was satisfactorily revised and now it can be accepted for publication at PLoS One.

Reviewers' comments:

Reviewer's Responses to Questions

**Comments to the Author**

1. If the authors have adequately addressed your comments raised in a previous round of review and you feel that this manuscript is now acceptable for publication, you may indicate that here to bypass the “Comments to the Author” section, enter your conflict of interest statement in the “Confidential to Editor” section, and submit your "Accept" recommendation.

Reviewer #1: All comments have been addressed

2. Is the manuscript technically sound, and do the data support the conclusions?

Reviewer #1: Yes

3. Has the statistical analysis been performed appropriately and rigorously? 

Reviewer #1: I Don't Know

4. Have the authors made all data underlying the findings in their manuscript fully available?

Reviewer #1: Yes

5. Is the manuscript presented in an intelligible fashion and written in standard English?

Reviewer #1: Yes

6. Review Comments to the Author

Reviewer #1: I recommend this manuscript to be accepted after the authors addressed all my comments and requests.

7. PLOS authors have the option to publish the peer review history of their article (what does this mean?). If published, this will include your full peer review and any attached files.

Reviewer #1: No

---

## [Editor Report · Acceptance letter]

27 Jun 2022

PONE-D-21-18008R2 

Mapping potential risks for the transmission of spotted fever rickettsiosis: the case study from the Rio de Janeiro state, Brazil. 

Dear Dr. RODRIGUES:

I'm pleased to inform you that your manuscript has been deemed suitable for publication in PLOS ONE. Congratulations! Your manuscript is now with our production department. 

Kind regards, 

on behalf of

Dr. Antonio Humberto Hamad Minervino 

Academic Editor

PLOS ONE